# Policy-Regret Minimization in Markov Games with Function Approximation

Thanh Nguyen-Tang [1]   Raman Arora [1]

## Abstract

We study policy-regret minimization problem in dynamically evolving environments, modeled as Markov games between a learner and a strategic, adaptive opponent. We propose a general algorithmic framework that achieves the optimal $\mathcal{O}(\sqrt{T})$ policy regret for a wide class of large-scale problems characterized by an Eluder-type condition–extending beyond the tabular settings of previous work. Importantly, our framework uncovers a simpler yet powerful algorithmic approach for handling reactive adversaries, demonstrating that leveraging opponent learning in such settings is key to attaining the optimal $\mathcal{O}(\sqrt{T})$ policy regret.

## 1. Introduction

In recent years, reinforcement learning (RL) has achieved remarkable success across a wide range of complex decision-making problems. Notable examples include mastering games such as Go (Silver et al., 2016; 2017; 2018), real-time strategy games like StarCraft II (Vinyals et al., 2019) and Dota (Berner et al., 2019), autonomous driving (Shalev-Shwartz et al., 2016), and socially interactive games such as hide-and-seek (Baker et al., 2019), capture-the-flag (Jaderberg et al., 2019), and poker (Texas hold'em) (Moravčík et al., 2017; Brown & Sandholm, 2018). Many of these settings can be naturally formulated as instances of multi-agent reinforcement learning (MARL), where multiple agents interact and learn to make decisions in a shared environment (Yang & Wang, 2020; Zhang et al., 2021).

Despite these empirical advances, the theoretical and algorithmic foundations of MARL remain relatively underdeveloped–particularly in scenarios where the learner faces *adaptive opponents* that adjust their strategies in response to the learner's behavior. Most prior work addresses this challenge through the lens of *external regret*, which

measures the gap between the learner's performance and that of the best fixed strategy in hindsight. However, external regret fails to capture the counterfactual nature of adaptivity: it overlooks how opponents might have reacted differently had the learner chosen a different strategy.

To address this limitation, *policy regret* (Arora et al., 2012b) has emerged as a more suitable notion. Unlike external regret, policy regret compares the learner's performance against the return they would have achieved by following an alternate strategy, thereby accounting for the adaptive behavior of opponents. Policy regret has been widely used to analyze learning against adaptive adversaries in online learning (Arora et al., 2012b; 2019) and repeated games (Arora et al., 2018), but only recently extended to the setting of multi-agent RL. In particular, Nguyen-Tang & Arora (2024a) initiated the study of policy regret in Markov games, establishing fundamental barriers and providing sufficient conditions for achieving sublinear policy regret in tabular environments. Their results hinge on two simplifying assumptions: small discrete state and action spaces, and *consistent behavior*–the adversary's response in a state depends solely on the learner's policy in that state.

However, these assumptions are restrictive. In realistic environments, an opponent's strategy in a particular state may depend on the learner's actions in other states. Moreover, in many real-world applications such as robotics and games, the state and action spaces can be exponentially large or even continuous and high-dimensional. making tabular guarantees impractical. While function approximation has proven effective in handling large-scale RL problems (Foster et al., 2021; Liu et al., 2022; Jin et al., 2021; Zhan et al., 2023; Liu et al., 2023; Chen et al., 2023; Zhong et al., 2022), its role in minimizing policy regret within MARL remains unexplored. This motivates the central question of our work:

**Can we design sample-efficient RL algorithms for policy regret minimization with function approximation?**

In this work, we introduce BOVL–the first algorithmic framework for policy-regret minimization in Markov games for a broad class of problems characterized by $\ell_2$-type Eluder coefficients. Our problem class subsumes the class of tabular problems studied in (Nguyen-Tang & Arora, 2024a) introduces a novel class of linear problems, extending the

---

[1]Department of Computer Science, Johns Hopkins University, Baltimore, MD, USA. Correspondence to: Raman Arora <arora@cs.jhu.edu>.

*Proceedings of the 42$^{nd}$ International Conference on Machine Learning*, Vancouver, Canada. PMLR 267, 2025. Copyright 2025 by the author(s).

| | (Nguyen-Tang & Arora, 2024a) | This work |
|---|---|---|
| **RL problems** | Two-player Markov games | Two-player Markov games |
| **Learner policies** | Deterministic | Deterministic/Stochastic |
| **Adversary behavior** | Consistent | Lipschitz |
| **Representation** | Tabular | General function approximation |
| **Expressivity** | Exact | Realizable, complete, and Eluder |
| **Policy regret** | $m\bar{V}\sqrt{HSAB}(SAB(H+\sqrt{S})+H^2)\sqrt{T/d^*}$ | $\bar{V}(m+H)\sqrt{d_E\gamma T}$ |

*Table 1.* A comparison between our work and (Nguyen-Tang & Arora, 2024a). Due to fundamental barriers in policy regret minimization for Markov games (Nguyen-Tang & Arora, 2024a), we only consider $m$-memory bounded and stationary adversaries for sample-efficient learning. Here, $T$ denotes the number of episodes, $m$ the memory of the adversary, $H$ the episode length, $S$, $A$ and $B$, the cardinalities of the state space, learner's action space, and adversary's action space, respectively, $d^*$ the minimum positive visitation probability, and $\gamma$, $d_E$ the covering-type and Eluder-type complexity measures of the function classes.

landscape of policy-regret minimization. Furthermore, our framework provides a significantly simpler algorithmic design while attaining tighter bounds when specialized to the tabular case. It also resolves three of the four open problems posed in (Nguyen-Tang & Arora, 2024a). Specifically,

- We identify a general class of problems characterized by $\ell_2$-type Eluder conditions (Condition 1 and Condition 2), which includes both the tabular case in (Nguyen-Tang & Arora, 2024a) and a new class of linear Markov games. Our framework also introduces a natural Lipschitz condition that models how adversaries respond similarly to similar policy sequences—an essential relaxation that overcomes the fundamental barriers identified by Nguyen-Tang & Arora (2024a, Theorem 3).

- We propose a generic algorithm, BOVL (Algorithm 1), tailored to this broader class. BOVL employs a batching strategy to handle adaptive, memory-bounded adversaries and constructs optimistic estimates of both value functions and adversary responses based on collected data. Compared to the layerwise exploration in Nguyen-Tang & Arora (2024a), BOVL is conceptually simpler and more amenable to large-scale settings.

- We show that BOVL attains a policy regret bound of $\bar{V}(m+H)\sqrt{d_E\gamma T}$, where $\bar{V}$ is the scale of the value functions, $m$ the memory of the adversary, $H$ the episode length, $d_E$ and $\gamma$ the Eluder-type and covering-type complexities of the function classes, respectively, and $T$ the total number of episodes. When instantiated to the tabular case, this bound improves significantly over Nguyen-Tang & Arora (2024a) by a factor of $\sqrt{\frac{HSA}{Bd^*}}$, where $d^*$ is the minimum positive visitation probability—which can be arbitrarily small in problems with large state and action spaces.

We summarize these key improvements in Table 1.

### 1.1. Technical Contribution

A core technical contribution of our work is a novel approach to handling the batching component of BOVL. Batching–originally inspired by Arora et al. (2012b)–divides the total time budget into equally sized intervals during which the algorithm commits to a fixed policy. This design is crucial for coping with adaptive (but memory-bounded) adversaries, as it allows the learner to observe consistent responses over each batch.

A key challenge in this setup is that the models trained at the beginning of a batch are not guaranteed to have small empirical loss on the data subsequently collected within that batch. Xiong et al. (2023) addressed this issue in the context of value function learning by showing that only a small number of batches exhibit large in-batch empirical error. Their analysis relies on a variant of $\ell_2$-type Eluder conditions. However, their technique applies only to empirical squared loss, which suffices for value function approximation but not for adversary learning in our setting, which involves empirical log-likelihood loss. The reason is that their approach requires a lower tail bound that is trivial for squared loss but remains unknown for log-likelihood.

To overcome this, we introduce a new approach that controls batches with large total variation loss rather than log-likelihood loss, thus avoiding reliance on unavailable lower-tail bounds (see Appendix F.2). Beyond our current setting, this technique naturally generalizes to obtain regret bounds with low switching costs in online density estimation problems with function approximation such as those studied in Liu et al. (2023).

## 2. Problem Setup and Preliminaries

We consider two-player Markov games (MGs), specified by the tuple $\mathcal{M} = (\mathscr{S}, \mathscr{A}, \mathscr{B}, H, \mathbb{P}, r)$, where $\mathscr{S}$ denotes the state space, $\mathscr{A}$ the learner's action space, and $\mathscr{B}$ the adversary's action space. The game proceeds over $H$ steps per episode. The reward function is given by $r = \{r_h\}_{h\in[H]}$,

where each $r_h : \mathscr{S} \times \mathscr{A} \times \mathscr{B} \to \mathbb{R}$. We define $[H] := \{1, \ldots, H\}$. The transition dynamics are given by $\mathbb{P} = \{\mathbb{P}_h\}_{h \in [H]}$, specifying the joint distribution over trajectories $s_{1:H}$ conditioned on action sequences $a_{1:H}$ and $b_{1:H}$:

$$\mathbb{P}(s_{1:H}|a_{1:H}, b_{1:H}) = \mathbb{P}_1(s_1) \prod_{h=2}^{H} \mathbb{P}_h(s_h|s_{h-1}, a_{h-1}, b_{h-1}).$$

Without loss of generality and for simplicity, we assume all episodes begin at the same fixed initial state $s_1$.

**Interaction Protocol.** At each episode, the learner and adversary interact over $H$ steps, starting from the fixed initial state $s_1$. At time step $h \in [H]$, the learner observes the current state $s_h$ and selects an action $a_h \in \mathscr{A}$, while the adversary selects an action $b_h \in \mathscr{B}$. The learner then observes $b_h$, receives a reward $r_h(s_h, a_h, b_h)$, and the environment transitions to the next state $s_{h+1} \sim \mathbb{P}_h(\cdot|s_h, a_h, b_h)$. The episode terminates after $H$ steps.

**Policies and Value Functions.** A (Markov) policy for the learner is denoted $\pi = \{\pi_h\}_{h \in [H]}$, where each $\pi_h(\cdot|s) \in \Delta(\mathscr{A})$ specifies a distribution over actions at step $h$ and state $s$. Similarly, the adversary's policy is $\mu = \{\mu_h\}_{h \in [H]}$, where $\mu_h(\cdot|s) \in \Delta(\mathscr{B})$. Here, $\Delta(\mathcal{X})$ denotes the set of probability measures over a measurable space $\mathcal{X}$. The value function at step $h$ and state $s$, under policy pair $(\pi, \mu)$, is defined as $V_h^{\pi,\mu}(s) := \mathbb{E}_{\pi,\mu}[\sum_{l=h}^{H} r_l(s_l, a_l, b_l)|s_h = s]$, where the expectation is taken over trajectories generated by the joint policy and transition dynamics. We also define the action-value function $Q_h^{\pi,\mu}(s, a, b) := \mathbb{E}_{\pi,\mu}[\sum_{l=h}^{H} r_l(s_l, a_l, b_l)|(s_h, a_h, b_h) = (s, a, b)]$. We assume that the learner and adversary select policies from restricted classes $\Pi_{\mathscr{A}}$ and $\Pi_{\mathscr{B}}$, respectively.

**Policy Norm.** To compare the similarity between two policies within $\Pi_{\mathscr{A}}$ or $\Pi_{\mathscr{B}}$, we define generic norms $\|\cdot\|_{\Pi_{\mathscr{A}}}$ and $\|\cdot\|_{\Pi_{\mathscr{B}}}$ on the corresponding policy spaces. Our results are stated in generality and do not depend on any specific choice of norm. However, to provide concrete examples:

- If policies are stochastic and the action space is discrete, one can use the $\ell_1/\ell_\infty$ norm:

$$\|\pi - \pi'\|_{\Pi_{\mathscr{A}}} = \max_{h \in [H]} \sup_{s \in \mathscr{S}} \|\pi_h(\cdot|s) - \pi'_h(\cdot|s)\|_1. \quad (1)$$

- In contrast, if $\Pi_{\mathscr{A}}$ contains only deterministic policies and $\mathscr{A}$ is continuous (e.g., in robotics), the norm above is ill-defined. In such cases, it is more appropriate to use:

$$\|\pi - \pi'\|_{\Pi_{\mathscr{A}}} = \max_{h \in [H]} \sup_{s \in \mathscr{S}} \|\pi_h(s) - \pi'_h(s)\|_{\mathscr{A}},$$

where $\pi_h : \mathscr{S} \to \mathscr{A}$ and $\|\cdot\|_{\mathscr{A}}$ is a norm on the continuous action space $\mathscr{A}$ (e.g., an $\ell_p$ norm).

**Additional Notation.** Let $f = (f_1, \ldots, f_H)$, where each $f_h : \mathscr{S} \times \mathscr{A} \times \mathscr{B} \to \mathbb{R}$. For any policy pair $(\pi, \mu)$, we define the expectation of $f_h$ under policies $\pi$ and $\mu$ as $f_h(s, \pi, \mu) := \mathbb{E}_{a \sim \pi_h(\cdot|s), b \sim \mu_h(\cdot|s)}[f_h(s, a, b)]$. We define the Bellman operator $\mathcal{T}_h^{\pi,\mu} : \mathbb{R}^{\mathscr{S} \times \mathscr{A} \times \mathscr{B}} \to \mathbb{R}^{\mathscr{S} \times \mathscr{A} \times \mathscr{B}}$ as:

$$[\mathbb{T}_h^{\pi,\mu} f_{h+1}](x) = r_h(x) + \langle \mathbb{P}_{h+1}(\cdot|x), f_{h+1}(\cdot, \pi, \mu)\rangle,$$

where $\langle\cdot, \cdot\rangle$ denotes the dot product between two vectors.

We denote the $p$-norm by $\|\cdot\|_p$, the maximum and minimum of two values by $x \vee y := \max\{x, y\}$ and $x \wedge y := \min\{x, y\}$, respectively. We write $f \lesssim g$ to indicate $f = \mathcal{O}(g)$, i.e., $f \leq cg$ for some absolute constant $c$. Similarly, we define the asymptotic relations $\gtrsim$ and $\asymp$ as the counterparts of $\geq$ and $=$ up to constant factors.

We also introduce the following shorthand notation:

- $x_h = (s_h, a_h, b_h)$, and $\mathcal{X} = \mathscr{S} \times \mathscr{A} \times \mathscr{B}$,
- $z_h := (s_h, a_h, b_h, r_h, s_{h+1})$, $\mathcal{Z} = \mathscr{S} \times \mathscr{A} \times \mathscr{B} \times \mathbb{R} \times \mathscr{S}$,
- $z_h^t = (s_h^t, a_h^t, b_h^t, r_h^t, s_{h+1}^t)$ for episode $t$.

Define

$$\mathcal{E}_h^{\pi,\mu}(f_h, f_{h+1})(x_h) = [\mathbb{T}_h^{\pi,\mu} f_{h+1} - f_h](x_h),$$
$$l_{\pi,\mu}(f_h, f_{h+1})(z_h) = (f_h(s_h, a_h, b_h) - r_h - f_{h+1}(s_{h+1}, \pi, \mu))^2$$
$$\Delta l_{\pi,\mu}(f_h, f_{h+1})(z_h) = l_{\pi,\mu}(f_h, f_{h+1})(z_h)$$
$$- l_{\pi,\mu}(\mathbb{T}_h^{\pi,\mu} f_{h+1}, f_{h+1})(z_h)$$

**Boundedness.** Without loss of generality and for simplicity, we assume that all rewards are bounded: $r_h \in [0, \bar{V}]$ for all $h \in [H]$, where $\bar{V}$ is a known constant.

**Adaptive Adversaries.** We consider adaptive adversaries, following the framework introduced by Arora et al. (2012b;a); Nguyen-Tang & Arora (2024a). Specifically, the adversary is allowed to adapt its behavior over time using arbitrary deterministic algorithms. Formally, at episode $t$, the adversary selects a policy $\mu^t$ using a (potentially unbounded) deterministic mapping $\mathfrak{A}_t : \Pi_{\mathscr{A}}^t \to \Pi_{\mathscr{B}}$, which maps the history of the learner's policies $\pi^1, \ldots, \pi^t$ to the adversary's response $\mu^t$.

This reactive model generalizes the canonical Stackelberg game setup (Von Stackelberg, 2010), where a "defender" (the learner) first commits to a strategy, and an "attacker" (the adversary) responds optimally, given knowledge of the defender's strategy.

**Policy regret minimization.** We measure the learner's performance using policy regret (Merhav et al., 2002; Arora et al., 2012b), which compares the learner's cumulative reward to that of the best fixed policy sequence in hindsight, accounting for the adaptive nature of the adversary. Formally, the policy regret after $T$ episodes is defined as:

$$\text{PR}(T) = \sup_{\pi \in \Pi_{\mathscr{A}}} \sum_{t=1}^{T} V_1^{\pi, \mathfrak{A}_t([\pi]^t)}(s_1) - V_1^{\pi^t, \mathfrak{A}_t(\pi^1, \ldots, \pi^t)}(s_1)$$

where $[\pi]^t$ denotes the repeated sequence $(\pi, \ldots, \pi)$ of length $t$, and $\mathfrak{A}_t([\pi]^t)$ is the adversary's policy if the learner had used $\pi$ in all past episodes.

# 3. Problems with Large State-Action Spaces

We begin by recalling the fundamental barriers to policy regret minimization in Markov games identified by Nguyen-Tang & Arora (2024a). These barriers hinge on two critical concepts: $m$-memory boundedness (Definition 1) and stationarity of the adversary (Definition 2). An adversary is said to be $m$-memory bounded if it only considers the learner's most recent $m$ policies. An adversary is stationary if its behavior does not vary over time.

**Definition 1** ($m$-memory bounded adversaries). *An adversary is said to be m-memory bounded for some $m \geq 0$, if for every $t$ and any policy sequence $\pi^1, \ldots, \pi^t$, we have:*

$$\mathfrak{A}_t(\pi^1, \ldots, \pi^t) = \mathfrak{A}_t(\pi^{1 \vee (t-m+1)}, \ldots, \pi^t).$$

**Definition 2** (Stationary adversaries). *An m-memory bounded adversary is said to be stationary if there exists a mapping $\mu^* : \Pi_{\mathscr{A}}^m \to \Pi_{\mathscr{B}}$ such that, for every $t$ and any policy sequence $\pi^1, \ldots, \pi^t$, we have:*

$$\mathfrak{A}_t(\pi^1, \ldots, \pi^t) = \mu^*(\pi^{1 \vee (t-m+1)}, \ldots, \pi^t).$$

Nguyen-Tang & Arora (2024a) show that policy regret minimization is not sample-efficient against adversaries that are either (i) unbounded in memory or (ii) memory-bounded but non-stationary.

Motivated by this impossibility result, we restrict our focus for the remainder of the paper to *memory-bounded and stationary adversaries* as formalized in Definition 2. Specifically, we assume that the adversary is governed by a fixed mapping $\mu^*$, and is $m$-memory bounded and stationary, where $m$ is known to the learner.

For convenience, we assume that the adversary employs Markov policies. Thus, we can write the joint distribution over adversary actions $b_{1:H}$ as:

$$\mu^*(b_{1:H} | s_{1:H}, \pi^{1 \vee (t-m+1):t}) = \prod_{h=1}^H \mu_h^*(b_h | s_h, \pi^{1 \vee (t-m+1):t}),$$

Whenever it is clear from the context, we also use the shorthand $\mu^*(\pi)$ to denote $\mu^*([\pi]^m)$.

## 3.1. Value Function Approximation

We consider the setting of large state spaces, where the state space $\mathscr{S}$ is exponentially large. In such cases, it is undesirable for the policy regret to scale polynomially with the number of states $|\mathscr{S}|$. To address this, it is common in practice–especially in deep RL (Mnih et al., 2015)–to use function approximation (e.g., neural networks) to estimate value functions. We adopt this approach in our work.

Formally, the learner is given a function class $\mathcal{F} = \mathcal{F}_1 \times \ldots \times \mathcal{F}_H$, where each $\mathcal{F}_h \subset \{\mathcal{X} \to [0, \bar{V}]\}$ provides a set of candidate functions to approximate $Q_h^{\pi, \mu^*([\pi]^m)}$ for any $\pi \in \Pi_{\mathscr{A}}$, with $\mu^*$ denoting the $m$-memory bounded and stationary adversary.

Learning in Markov games remains extremely challenging without additional expressivity assumptions on the function class. We adopt the standard assumption of Bellman completeness.

**Assumption 3.1** (Bellman completeness). *For all $\pi \in \Pi_{\mathscr{A}}, f \in \mathcal{F}$, and $h \in [H]$, we have $\mathbb{T}_h^{\pi, \mu^*([\pi]^m)} f_{h+1} \in \mathcal{F}_h$.*

Bellman completeness requires the function class to be closed under the Bellman operator. It also implies realizability, i.e., $Q_h^{\pi, \mu^*([\pi]^m)} \in \mathcal{F}_h, \forall (\pi, h) \in \Pi_{\mathscr{A}} \times [H]$. This assumption is widely used as a sufficient condition for sample-efficient RL with function approximation (see, e.g., Jin et al. (2021)).

**Covering number.** When the function class $\mathcal{F}$ and policy class $\Pi_{\mathscr{A}}$ are finite, their statistical complexity can be measured via their log cardinalities, $\log |\mathcal{F}|$ and $\log |\Pi_{\mathscr{A}}|$. For infinite classes, we use the standard notion of $\varepsilon$-covering number.

**Definition 3** ($\epsilon$-covering). *For any $\epsilon > 0$, the $\epsilon$-covering number of a pseudometric space $(\mathcal{X}, d)$, denoted by $N(\epsilon; \mathcal{X}, d)$, is the smallest integer $n$ such that there exists a subset $\mathcal{X}_\epsilon \subseteq \mathcal{X}$ with $|\mathcal{X}_\epsilon| = n$ and $\sup_{x \in \mathcal{X}} \inf_{x' \in \mathcal{X}_\epsilon} d(x, x') \leq \epsilon$.*

We apply the covering numbers to the spaces $(\mathcal{F}, \|\cdot\|_\infty)$ and $(\Pi_{\mathscr{A}}, \|\cdot\|_{\Pi_{\mathscr{A}}})$, where $\|f - g\|_\infty := \max_h \|f_h - g_h\|_\infty$. For notational simplicity, we omit the metric dependence and write $N_{\mathcal{F}}(\epsilon)$ and $N_{\Pi_{\mathscr{A}}}(\epsilon)$, instead of $N(\epsilon; \mathcal{F}, \|\cdot\|_\infty)$ and $N(\epsilon; \Pi_{\mathscr{A}}, \|\cdot\|_{\Pi_{\mathscr{A}}})$, respectively.

**Example 3.1** (Linear function approximation). *Consider a linear Markov game with feature map $\phi : \mathcal{X} \to \mathbb{R}^{d_{in}}$, i.e., $\forall h \in [H]$, the transition and reward functions are given by $\mathbb{P}_h(s'|x) = \langle \phi(x), \nu_h(s') \rangle$ and $r_h(x) = \langle \phi(x), \theta_h \rangle$, for some functions $\nu_h(\cdot)$ and parameters $\theta_h \in \mathbb{R}^{d_{in}}$ (Jin et al., 2023). The function class $\mathcal{F} \subset \{x \mapsto \langle \phi(x), w \rangle : w \in \mathbb{R}^{d_{in}}\}$.*

**Lemma 3.1.** *In Example 3.1, $\log N_{\mathcal{F}}(\epsilon) = \mathcal{O}(d_{in} H \log(1/\epsilon))$.*

**Lipschitzness assumption.** Finally, we assume that functions in $\mathcal{F}$ are Lipschitz with respect to both the learner's and adversary's policies.

**Assumption 3.2.** *(Lipschitzness for Value Functions) There exists a constant $Lip_Q$ such that for all $\pi, \pi' \in \Pi_{\mathscr{A}}$,*

$\mu, \mu' \in \Pi_{\mathcal{B}}$, $f \in \mathcal{F}$ and $(s, h) \in (\mathcal{S}, [H])$,

$$|f_h(s, \pi, \mu) - f_h(s, \pi', \mu)| \leq Lip_Q \cdot \|\pi - \pi'\|_{\Pi_{\mathcal{A}}},$$
$$|f_h(s, \pi, \mu) - f_h(s, \pi, \mu')| \leq Lip_Q \cdot \|\mu - \mu'\|_{\Pi_{\mathcal{B}}}.$$

For example, if we define $\|\cdot\|_{\Pi_{\mathcal{A}}}$ and $\|\cdot\|_{\Pi_{\mathcal{B}}}$ using the $\ell_1/\ell_\infty$ as in Equation (1) norm, then $\text{Lip}_Q \leq \bar{V}$.

### 3.2. Opponent Function Approximation

Nguyen-Tang & Arora (2024a, Theorem 3) show that achieving sublinear policy regret is impossible–even against $m$-memory bounded and stationary adversaries–if there is no constraint on how the adversary responds to two similar sequences of learner policies. To address this issue, they propose a sufficient condition for learnability, namely *consistency*: if the learner plays two sequences of policies that agree at certain states $s$ and steps $h$, then a consistent adversary should respond with policies that also agree at those same states and steps.

**Definition 4** (Consistent adversaries (Nguyen-Tang & Arora, 2024a))**.** *An $m$-memory bounded and stationary adversary is said to be consistent if, for any two sequences of learner's policies $\pi^1, \ldots, \pi^m$ and $\nu^1, \ldots, \nu^m$, and any $(s, h) \in \mathcal{S} \times [H]$, the following holds:* ***if*** *$\pi_h^i(\cdot|s) = \nu_h^i(\cdot|s), \forall i \in [m]$,* ***then*** *$\mu_h^*(\cdot|s, \pi^1, \ldots, \pi^m) = \mu_h^*(\cdot|s, \nu^1, \ldots, \nu^m)$.*

In essence, the adversary's response at step $h$ and state $s$ depends only on how the learner's policies behave at that state and step. While consistency enables learnability in tabular Markov games by overcoming the fundamental hardness barrier, it significantly limits the class of admissible adversaries and does not generalize to large state spaces.

To overcome this limitation, we instead assume that the adversary's response is Lipschitz in the learner's policy:

**Assumption 3.3** (**Lipschitz Adversary**)**.** *There exists a constant $Lip_{Adv}$ such that for all $\pi, \pi' \in \Pi_{\mathcal{A}}$, we have*

$$\|\mu^*([\pi]^m) - \mu^*([\pi']^m)\|_{\Pi_{\mathcal{B}}} \leq Lip_{Adv} \cdot \|\pi - \pi'\|_{\Pi_{\mathcal{A}}}.$$

This assumption ensures that similar learner policies yield similar adversary responses, under the respective policy norms. It generalizes the consistency assumption: for instance, even if $\pi_h = \pi'_h$ for some $h \in [H]$, a $\text{Lip}_{\text{Adv}}$-Lipschitz adversary need not satisfy $\mu_h^*(\pi) = \mu_h^*(\pi')$, as required under consistency. When $\Pi_{\mathcal{A}}$ contains only deterministic policies, a consistent adversary is 1-Lipschitz.

**Behavior Model Class.** We assume that the learner has access to a known model class of adversary behaviors that contains the true adversary. Specifically, the learner is given a behavior model class $\Psi = \Psi_1 \times \cdots \times \Psi_H$, where each $\mu_h \in \Psi_h$ maps $\Pi_{\mathcal{A}} \times \mathcal{S}$ to distributions over $\mathcal{B}$. We assume realizability:

**Assumption 3.4** (**Adversary realizability**)**.** *The adversary's response function satisfies $\mu^*([\cdot]^m) \in \Psi$.*

**Remark 1.** *We only model how $\mu^*$ responds to a repeated sequence $[\pi]^m = (\pi, \ldots, \pi)$, as this is the benchmark used in defining policy regret. No assumptions are made about how $\mu^*$ responds to arbitrary sequences of differing policies.*

Following standard practice in maximum likelihood estimation (MLE) analysis (Geer, 2000), we measure the complexity of $\Psi$ using the bracketing number:

**Definition 5** (Bracketing number)**.** *A collection $\{[l_i, u_i], i \in [N]\}$ of function pairs $l_i, u_i : \Pi_{\mathcal{A}} \times \mathcal{S} \times \mathcal{B} \to \mathbb{R}_+$ is said to be an $\epsilon$-bracketing cover of $\Psi_h$ if*

- $\max_{i \in [N]} \sup_{\pi \in \Pi_{\mathcal{A}}} \sup_{s \in \mathcal{S}} \|u_i(\pi, s, \cdot)\|_1 \leq \mathcal{O}(1)$;

- *For every $\mu_h \in \Psi_h$ there exists $i \in [N]$ such that $l_i(\pi, s, b) \leq \mu_h(b|s, \pi) \leq u_i(\pi, s, b)$ for all $(\pi, s, b)$;*

- $\max_{i \in [N]} \sup_{\pi \in \Pi_{\mathcal{A}}} \sup_{s_h \in \mathcal{S}} \|l_i(\pi, s_h, \cdot) - u_i(\pi, s_h, \cdot)\|_1 \leq \epsilon.$

*The smallest such $N$, denoted by $N_{\Psi_h}^{[]}(\epsilon)$, is called the $\epsilon$-bracketing number of $\Psi_h$. The $\epsilon$-bracketing number of $\Psi$, denoted by $N_\Psi^{[]}(\epsilon)$, is defined as $N_\Psi^{[]}(\epsilon) = \max_{h \in [H]} N_{\Psi_h}^{[]}(\epsilon)$.*

The bracketing number gives stronger control over model complexity than the covering number and is commonly used in model-based RL (Liu et al., 2022; 2023). Unlike those works, our definition does not assume a specific factorization of $\mu_h(\pi)$.

**Running Example.** We illustrate these ideas with a linear adversary response model:

**Example 3.2** (**Linear Response**)**.** *For all $(s, h, \pi)$,*

$$\mu_h^*(\cdot|s, [\pi]^m) = \langle \Phi^*, w_{hs}^\pi \rangle$$

*where $\Phi^* : \mathbb{R}^{d_{adv}} \to \mathbb{R}_+^{\mathcal{B}}$ maps $v \in \mathbb{R}^{d_{adv}}$ to $\langle \Phi^*, v \rangle$, and $w_{hs}^\pi \in \mathbb{R}^{d_{adv}}$ that depends on $\pi, h, s$ with $\|w_{hs}^\pi\|_1 = \mathcal{O}(1)$.*

In the tabular setting of Nguyen-Tang & Arora (2024a) with consistent adversary and $\Pi_{\mathcal{A}}$ including only deterministic policies, we recover this form with $d_{adv} = HSA$ and $w_{hs}^\pi$ being the one-hot vector corresponding to $(h, s, \pi_h(s))$.

**Lemma 3.2.** *Assume the linear response in Example 3.2. Let $\Psi = \Psi_1 \times \ldots \times \Psi_H$ where each $\Psi_h$ is the set of $\mu_h$*

*such that* $\mu_h(\cdot|s, \pi) = \langle \Phi, w_{hs}^\pi \rangle$ *with* $\|\Phi\|_\infty = \mathcal{O}(1)$. *Then the bracketing number of* $\Psi$ *can be bounded as:*

$$\log N_\Psi^{[]}(\epsilon) = \mathcal{O}\left(d_{adv} B \log(B/\epsilon)\right).$$

The proof is given in Appendix G.

## 4. Algorithmic Framework

In this section, we introduce a simple yet general algorithm–**B**atching and **O**ptimism based on **V**alue and **L**ikelihood fitting (BOVL)–for policy regret minimization with function approximation. The pseudocode is provided in Algorithm 1.

BOVL combines both value-based and model-based learning: the former for learning value functions, and the latter for modeling the adversary's behavior. It takes as input the value function class $\mathcal{F}$, the adversary model class $\Psi$, the policy class $\Pi_{\mathscr{A}}$, the number of effective episodes $T$[1], and the number of batches $K$.

The algorithm proceeds as follows:

- **Optimistic planning** (Line 3): At each episode $t \in [T]$, BOVL computes an optimistic policy $\pi^t$ that maximizes the expected return under the most optimistic combination of a value function and an adversary model.
- **Data collection** (Line 4): The learner executes policy $\pi^t$ to collect a trajectory $\tau^t$, with the adversary responding based on the most recent $m$ policies of the learner.
- **Periodic confidence set updates** (Lines 5–8): Every $\lfloor T/K \rfloor$ episodes, the algorithm updates its confidence sets $\mathcal{F}^t(\cdot, \cdot)$ and $\Psi^t$ using all collected data. In between updates, the confidence sets remain fixed (Line 9).

A core component of BOVL is the construction of confidence sets $\mathcal{F}^t(\cdot, \cdot)$ and $\Psi^t$. The set $\mathcal{F}^t(\pi, \nu)$ includes all value functions $f \in \mathcal{F}$ that approximately explain the data collected so far, up to an error $\alpha$, in terms of squared loss as a proxy for the Bellman error under the policy pair $(\pi, \nu)$. This construction is rather standard in RL and is similar to that in the GOLF algorithm of Liu et al. (2023). The set $\Psi^t$ includes all models $\mu \in \Psi$ whose log-likelihood over the collected data is within $\beta$ of the best possible model in $\Psi$. This can be viewed as a relaxation of maximum likelihood estimation: when $\beta = 0$, $\Psi^t$ reduces to a singleton containing the MLE.

The key design in BOVL for handling $m$-memory bounded adversaries is its batching strategy. The algorithm partitions the total episode budget $T + (m-1)K$ into evenly spaced intervals (batches) of size $m - 1 + \lfloor T/K \rfloor$, and holds the policy fixed throughout each batch. The confidence sets– and therefore the policy–are only updated at the start of each

batch. This structure allows the learner to observe and infer the adversary's behavior in response to repeated executions of the same policy. Notably, Line 6 implements a "warm-up" phase where the learner executes $\pi^t$ for $m - 1$ episodes (without collecting data) to ensure that the adversary's response stabilizes to the current policy.

**Remark 2.** *The tri-level optimization in Line 3 generally cannot be solved efficiently, and our analysis is primarily theoretical.*

**Comparison with the algorithm by (Nguyen-Tang & Arora, 2024a) for tabular MGs.** Since our work resolves several open questions posed by Nguyen-Tang & Arora (2024a), it is useful to contrast the algorithmic design principles of both works.

- **Similarities**: Both algorithms employ batching to address $m$-memory bounded adversaries. They also share the high-level idea of jointly learning the environment and the adversary through optimism over value functions and opponent models.

- **Differences**: Our algorithm handles large state and action spaces through function approximation, whereas Nguyen-Tang & Arora (2024a) is limited to tabular Markov games. Their algorithm uses a complex layerwise exploration mechanism that requires full coverage of the state-action space at every time step $h \in [H]$. This involves marking infrequent transitions and truncating reward functions—designs that are fragile and impractical in large or continuous domains. Furthermore, their method depends on knowledge of the minimum nonzero visitation probability, a strong assumption that may not hold in practice. In contrast, BOVL uses more modular and interpretable components—standard squared-loss fitting for value functions and log-likelihood based updates for opponent modeling—leading to a significantly simpler and more general algorithm.

## 5. Theoretical Guarantees

In this section, we establish theoretical guarantees for our algorithm BOVL. Specifically, we introduce a general sufficient condition, called the $\ell_2$-type Eluder condition, and prove that for any Markov game satisfying this condition, BOVL achieves $\tilde{\mathcal{O}}(\sqrt{T})$ policy regret.

### 5.1. $\ell_2$-type Eluder Conditions

The Eluder condition is a structural complexity measure inspired by the pigeonhole principle and the elliptical potential lemma, which have been widely used in tabular MDPs (Jin et al., 2018) and linear MDPs (Jin et al., 2023). Variants of Eluder conditions have become a standard tool for analyzing optimistic exploration algorithms in both bandit and RL

---

[1]BOVL runs for $T + (m-1)K$ episodes due to Line 6.

---

**Algorithm 1** $\text{BOVL}(\mathcal{F}, \Psi, \Pi_{\mathscr{A}}, T, K)$ – **B**atching and **O**ptimism based on **V**alue and **L**ikelihood fitting

---

1: **Initialize**: $\Psi^0 = \Psi$, and $\mathcal{F}^0(\pi, \mu(\pi)) = \mathcal{F}, \forall (\pi, \mu) \in \Pi_{\mathscr{A}} \times \Psi$,

$$\alpha \asymp \bar{V}^2 \log(2N_{\mathcal{F}}(1/T)N_{\Pi_{\mathscr{A}}}(1/T)TH/\delta) + \bar{V}\text{Lip}_Q(1 + \text{Lip}_{\text{Adv}}),$$
$$\beta \asymp \log\left(N_{\Psi}^{[]}(1/T)TH/\delta\right).$$

2: **for** t = 1, ..., T **do**
3:     $\pi^t = \arg\max\limits_{\pi \in \Pi_{\mathscr{A}}} \max\limits_{\mu \in \Psi^{t-1}} \max\limits_{f \in \mathcal{F}^{t-1}(\pi, \mu(\pi))} f_1(s_1, \pi, \mu(\pi))$
4:     Execute $\pi^t$ to collect a trajectory $\tau^t = \{(s_h^t, a_h^t, b_h^t, r_h^t)\}_{h \in [H]}$ (the adversary responds to the last $m$ policies of the learner, including $\pi^t$)
5:     **if** $t = t_j := j\lfloor T/K \rfloor + 1$ for some integer $j$ **then**
6:         Execute $\pi^t$ for $m - 1$ consecutive episodes (and collect nothing)
7:         Update the confidence sets $\mathcal{F}^t(\cdot, \cdot)$ and $\Psi^t$ as follows:

$$\mathcal{F}^t(\pi, \nu) := \left\{ f \in \mathcal{F} : \sum_{i=1}^t l_{\pi,\nu}(f_h, f_{h+1})(z_h^i) - \inf_{g_h \in \mathcal{F}_h} \sum_{i=1}^t l_{\pi,\nu}(g_h, f_{h+1})(z_h^i) \leq \alpha, \forall h \in [H] \right\}$$

$$\Psi^t := \left\{ \mu \in \Psi : \sup_{\mu_h' \in \Psi_h} \sum_{i=1}^t \log \frac{\mu_h'(b_h^i|s_h^i, \pi^i)}{\mu_h(b_h^i|s_h^i, \pi^i)} \leq \beta, \forall h \in [H] \right\}$$

where $l_{\pi,\nu}(f_h, f_{h+1})(s, a, b, r, s') = (f_h(s, a, b) - r - f_{h+1}(s', \pi, \nu))^2$ and $z_h^i = (s_h^i, a_h^i, b_h^i, r_h^i, s_{h+1}^i)$.
8:     **else**
9:         $\mathcal{F}^t(\cdot, \cdot) \equiv \mathcal{F}^{t-1}(\cdot, \cdot)$ and $\Psi^t \equiv \Psi^{t-1}$
10:     **end if**
11: **end for**

---

settings (e.g., Russo & Van Roy (2013); Liu et al. (2022); Jin et al. (2021); Zhan et al. (2023); Liu et al. (2023); Chen et al. (2023); Zhong et al. (2022)).

In our case, we extend the Eluder condition to two components of learning: value function approximation and opponent modeling. Moreover, due to the batching nature of BOVL, we require a slightly stronger version known as the $\ell_2$-type Eluder condition—first introduced in the context of online learning with limited adaptivity by Xiong et al. (2023). This condition is critical for controlling errors within each batch where confidence sets are not updated.

To highlight the distinction between the standard and $\ell_2$-type Eluder conditions, we begin with a simple example:

**Example 5.1** (Standard vs. $\ell_2$-type Eluder Conditions). *Let $\Phi$ and $\mathcal{Y}$ be vector spaces. Consider sequences $\{\phi_i\}$ in $\Phi$ and $\{y_i\}$ in $\mathcal{Y}$, and let $\lambda > 0$.*

*The standard Eluder condition characterizes the complexity of $(\Phi, \mathcal{Y})$ by the smallest $d$ such that if $\sum_{i=1}^{t-1} \langle \phi_t, y_i \rangle^2 \leq \lambda$ for all $t \in [T]$, then the cumulative sum of inner products is bounded: $\sum_{i=1}^t \langle \phi_i, y_i \rangle \leq \sqrt{dT\lambda}$.*

*The $\ell_2$-type Eluder condition strengthens this conclusion by bounding the sum of squared inner products, requiring that $\sum_{i=1}^t \langle \phi_i, y_i \rangle^2 \leq d\lambda \log T$.*

*Clearly, the $\ell_2$-type condition is stronger. In fact, it implies the standard one via Cauchy–Schwarz: $\sum_{i=1}^t \langle \phi_i, y_i \rangle \leq \sqrt{T \sum_{i=1}^t \langle \phi_i, y_i \rangle^2} \leq \sqrt{dT\lambda \log T}$.*

We now formally define the $\ell_2$-type Eluder conditions for value functions and adversary models.

**Condition 1** ($\ell_2$-type Eluder coefficient for value function class). *The $\ell_2$-type Eluder coefficient of $(\mathcal{F}, \Pi_{\mathscr{A}})$, denoted by $dim_E(\mathcal{F}, \Pi_{\mathscr{A}})$, is the smallest $d$ such that for any $T \in \mathbb{N}$ and any sequences $\{f^t\}_{t \in [T]} \subset \mathcal{F}$, $\{\pi^t\}_{t \in [T]} \subset \Pi_{\mathscr{A}}$ and $\{x_h^t\}_{t \in [T]} \subset \mathscr{S} \times \mathscr{A} \times \mathscr{B}$ and for any $\lambda > 0$, the following holds:*

$$if \quad \sum_{i=1}^{t-1} \mathcal{E}_h^{\pi^t, \mu^*(\pi^t)}(f_h^t, f_{h+1}^t)(x_h^t)^2 \leq \lambda \quad \forall t \in [T],$$

$$then \quad \sum_{i=1}^t \mathcal{E}_h^{\pi^i, \mu^*(\pi^i)}(f_h^i, f_{h+1}^i)(x_h^i)^2 \leq \mathcal{O}(d\lambda \log t).$$

This condition states that small cumulative in-sample Bellman errors imply small cumulative in-distribution Bellman errors, where the bound depends on $dim_E(\mathcal{F}, \Pi_{\mathscr{A}})$.

**Lemma 5.1.** *The Eluder coefficient for the linear model in Example 3.1 satisfies $dim_E(\mathcal{F}, \Pi_{\mathscr{A}}) = \mathcal{O}(d_{in})$ for any $\Pi_{\mathscr{A}}$.*

| | Function Approximation | Policy Regret |
|---|---|---|
| **BOVL (Ours)** | General | $\bar{V}(m+H)\sqrt{d_E\gamma T}$ |
| | Linear | $\bar{V}(m+H)(d_{\text{in}} \vee d_{\text{adv}})\sqrt{(H+B)T}$ |
| | Tabular | $\bar{V}(m+H)SA\sqrt{(H+B)^3T}$ |
| **APE-OVE** (Nguyen-Tang & Arora, 2024a) | Tabular | $m\bar{V}\sqrt{HSAB}(SAB(H+\sqrt{S})+H^2)\sqrt{\frac{T}{d^*}}$ |

*Table 2.* A summary of our main result and its instantiation to the linear case and the tabular case.

**Condition 2** ($\ell_2$-type Eluder coefficient for adversary class). *Let $dim_E(\Psi, \Pi_{\mathscr{A}})$ denote the $\ell_2$-type Eluder coefficient of the adversary class $\Psi$. It is the smallest $d$ such that for any $T \in \mathbb{N}$, any sequences $\{\mu_1^t\}_{t\in[T]} \subset \Psi_1$ and $\{\pi^t\}_{t\in[T]} \subset \Pi_{\mathscr{A}}$, and any $\lambda > 0$, the following holds:*

$$\text{if} \quad \sum_{i=1}^{t-1} \|\mu_1^t(\pi^i) - \mu_1^*(\pi^i)\|_1^2 \le \lambda \quad \forall t \in [T],$$

$$\text{then} \quad \sum_{i=1}^{t} \|\mu_1^i(\pi^i) - \mu_1^*(\pi^i)\|_1^2 \le \mathcal{O}(d\lambda \log t).$$

This condition extends Eluder-type reasoning to adversary models using total variation (TV) distance. Similar assumptions have been employed in the optimistic MLE literature (e.g., Liu et al. (2023), Condition 3.1).

**Lemma 5.2.** *Consider the linear response model in Example 3.2 with dimension $d_{adv}$. Then $dim_E(\Psi, \Pi_{\mathscr{A}}) = \mathcal{O}(d_{adv})$ for any $\Pi_{\mathscr{A}}$.*

## 5.2. Main Result

We now present the main result, which provides a theoretical guarantee on the policy regret of BOVL.

**Theorem 1.** *Fix any $\delta \in (0,1)$. Under Assumptions 3.1, 3.2, 3.3 and 3.4, and Conditions 1 and 2, if we set $K \asymp \sqrt{\frac{T \cdot d_E \cdot \log^3 T}{\gamma}}$ in BOVL, then with probability at least $1 - \delta$,*

$$PR(T) = \mathcal{O}(\bar{V}(H+m)\sqrt{d_E\gamma T \log^3 T}),$$

*where $\gamma = \frac{\alpha}{\bar{V}^2} \vee \beta$, $d_E = dim_E(\mathcal{F}, \Pi_{\mathscr{A}}) \vee dim_E(\Psi, \Pi_{\mathscr{A}})$.*

This result shows that the policy regret of BOVL is independent of the cardinalities of the state space $\mathscr{S}$ and action spaces $\mathscr{A}, \mathscr{B}$, making BOVL suitable for large-scale problems, including those with continuous high-dimensional actions. We instantiate this result for two cases.

For linear value functions (see Example 3.1) and linear adversary models (see Example 3.2), we can bound $\gamma = \mathcal{O}((Hd_{\text{in}} \vee Bd_{\text{adv}}) \log(BT))$ (by Lemma 3.1 and Lemma 3.2) and $d_E = \mathcal{O}(d_{\text{in}} \vee d_{\text{adv}})$ (by Lemma 5.1 and Lemma 5.2).

The tabular case studied in Nguyen-Tang & Arora (2024a) with consistent adversaries corresponds to the linear case with $d_{\text{in}} = SAB$ and $d_{\text{adv}} = HSA$.

We summarize these instantiated bounds in Table 2. Even in the tabular case, our policy regret bound improves over Nguyen-Tang & Arora (2024a) by a factor of $\sqrt{\frac{HSA}{Bd^*}}$, which can be significantly large when $d^*$ is small, as is often the case in finite but large-state problems.

## 5.3. Sketch Proof of the Main Theorem

We outline the main steps in proving the policy regret bound for BOVL. The full proof appears in Appendix A.

**Step 1: Optimism and Error Decomposition.** We begin by showing that, with high probability, the confidence sets $\Psi^{t-1}$ contain the true adversary model $\mu^*$ and $\mathcal{F}^{t-1}(\pi, \nu)$ contains the value function $Q^{\pi,\nu}$ for all $\pi, \nu$ and $t \in [T]$ (see Lemma D.1). This follows from martingale concentration and the construction of confidence radii $\alpha$ and $\beta$ in BOVL.

Let

$$(f^t, \mu^t) = \underset{f \in \mathcal{F}^{t-1}(\pi^t, \mu(\pi^t)), \mu \in \Psi^{t-1}}{\arg\max} f_1(s_1, \pi^t, \mu(\pi^t)).$$

Each time the algorithm switches policies or plays a policy for fewer than $m$ consecutive episodes, it may incur up to $\mathcal{O}(\bar{V})$ additional regret. Since such events happen at most $mK$ times (due to $K$ policy switches and $(m-1)K$ instances of "immature" plays), the total regret can be decomposed as:

$$PR(T + (m-1)K) \lesssim mK\bar{V} + \bar{V}\sum_{t=1}^{T} \|\mu_1^t(\pi^t) - \mu_1^*(\pi^t)\|_1$$

$$+ \sum_{t=1}^{T}\sum_{h=1}^{H} \mathcal{E}_h^{\pi^t, \mu^*(\pi^t)}(f_h^t, f_{h+1}^t)(x_h^t). \quad (2)$$

**Step 2: Leveraging the Confidence Set Design** By design, for each batch $j \in \{0, \ldots, K-1\}$, construction of the confidence sets in Line 7 forces $f^{t_j+1}$ and $\pi^{t_j+1}$ to have small squared loss (a proxy for Bellman error). Using standard martingale concentration inequalities (Lemma E.1), we can show that, with high probability

$$\sum_{i=1}^{t_j} \mathcal{E}_h^{\pi^{t_j+1}, \mu^*(\pi^{t_j+1})}(f_h^{t_j+1}, f_{h+1}^{t_j+1})(x_h^i)^2 \lesssim \alpha.$$

Because confidence sets are only updated at the start of each batch, we cannot guarantee that the total empirical squared Bellman error will be small for all episodes within the batch. However, by the $\ell_2$-type Eluder condition (Condition 1), the number of batches where this fails is small. In particular, the cardinality of the set $\mathcal{K}^s$ of all batches with large total empirical squared is bounded as (Lemma F.2):

$$|\mathcal{K}^s| \lesssim \dim_E(\mathcal{F}, \Pi_{\mathscr{A}}) \log T \log(T\bar{V}^2). \qquad (3)$$

Then, for all episodes $t$ in any batch in $\{0, \ldots, K-1\}/\mathcal{K}^s$:

$$\sum_{i=1}^{t-1} \mathcal{E}_h^{\pi^t, \mu^*(\pi^t)}(f_h^t, f_{h+1}^t)(x_h^i)^2 \lesssim \alpha. \qquad (4)$$

Similarly, by Condition 2, the number of batches with large total variation error is also small (Lemma F.4):

$$|\mathcal{K}^{tv}| \lesssim \dim_E(\Psi, \Pi_{\mathscr{A}}) \log^2 T, \qquad (5)$$

and for all $t$ in batches indexed by $\{0, \ldots, K-1\}/\mathcal{K}^{tv}$:

$$\sum_{i=1}^{t-1} \|\mu_1^t(\pi^i) - \mu_1^*(\pi^i)\|_1^2 \lesssim \beta. \qquad (6)$$

**Step 3: From Conditions to Final Regret Bound.** Using the $\ell_2$-type Eluder conditions, we can now relate the bounds above to the regret decomposition in Eq. (2). In particular, by Cauchy–Schwarz:

$$\sum_{t \in \{0, \ldots, K-1\}/\mathcal{K}^s} \mathcal{E}_h^{\pi^t, \mu^*(\pi^t)}(f_h^t, f_{h+1}^t)(x_h^t)$$
$$\lesssim \sqrt{T \dim_E(\mathcal{F}, \Pi_{\mathscr{A}}) \alpha \log T}. \qquad (7)$$

A similar chain of arguments establish that

$$\sum_{t \in \{0, \ldots, K-1\}/\mathcal{K}^{tv}} \|\mu_1^t(\pi^t) - \mu_1^*(\pi^t)\|_1$$
$$\lesssim \sqrt{T \dim_E(\Psi, \Pi_{\mathscr{A}}) \beta \log T}. \qquad (8)$$

Combining Eqs. (2), (7), (8), and the bounds on $|\mathcal{K}^s|$ and $|\mathcal{K}^{tv}|$, we obtain the desired $\tilde{\mathcal{O}}(\sqrt{T})$ policy regret.

## 6. Conclusion and Discussion

In this work, we develop the first general algorithmic framework for policy regret minimization in Markov games with function approximation, addressing a fundamental open problem in sequential decision-making under strategic interaction. Unlike prior works that are restricted to tabular settings or rely on strong assumptions such as consistency of adversaries, our framework handles large-scale and continuous environments through function approximation. Our approach achieves an optimal $\mathcal{O}(\sqrt{T})$ policy regret bound under natural assumptions, notably $\ell_2$-type Eluder conditions that jointly characterize the complexity of the value function and opponent model classes.

A key technical insight in our design is the batching mechanism that enables the learner to commit to policies over multiple rounds, which is critical for learning in the presence of reactive adversaries. To overcome the challenge that confidence sets are not updated within a batch, we introduce a novel analysis technique that quantifies the number of batches with large in-sample error using Eluder-type arguments. This allows us to tightly control the regret contributions from both value estimation and opponent modeling. In contrast to previous work, which only handles squared loss for value estimation, our method supports total variation loss and bypasses the need for lower tail bounds on log-likelihood, which are unknown.

We also demonstrate that our regret bound remains independent of the cardinality of the state and action spaces, making our algorithm suitable for continuous high-dimensional domains. When specialized to linear or tabular models, our bounds not only recover but also strictly improve upon those of prior work, such as Nguyen-Tang & Arora (2024a), with significant gains in the tabular case.

While our framework provides a principled foundation for learning in Markov games with function approximation, several important directions remain open. First, extending our guarantees to adversaries with unbounded memory or greater adaptivity would broaden the scope of our framework. Second, it would be valuable to explore alternative complexity measures that can relax or replace the current Lipschitz assumptions. Third, designing adaptive batching schemes could further improve empirical performance, especially in non-stationary settings. Fourth, validating the framework on large-scale benchmarks would help bridge the gap between theory and practice. Finally, generalizing the algorithm to cooperative or aligned multi-agent scenarios could open new avenues for applying policy regret minimization to collaborative decision-making.

## Impact Statement

This paper presents work whose goal is to advance the field of Machine Learning. There are many potential societal consequences of our work, none which we feel must be specifically highlighted here.

## Acknowledgements

This research was supported, in part, by the DARPA GARD award HR00112020004 and NSF CAREER award IIS-1943251.

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

## A. Full Proof of Main Theorem

We restate Theorem 1 with more details in the following theorem.

**Theorem 2.** *Fix any $\delta \in (0, 1)$. Under Assumptions 3.1, 3.2, 3.3, 3.4, and Conditions 1, 2, with probability at least $1 - \delta$, the policy regret of BOVL can be bounded as*

$$PR(T + (m-1)K) \lesssim mK\bar{V} + H\sqrt{\alpha \cdot T \cdot dim_E(\mathcal{F}, \Pi_{\mathscr{A}}) \cdot \log T} + \bar{V}\sqrt{\beta \cdot T \cdot dim_E(\Psi, \Pi_{\mathscr{A}}) \cdot \log T}$$
$$+ H\bar{V}\frac{T}{K} \cdot dim_E(\mathcal{F}, \Pi_{\mathscr{A}}) \cdot \log^2 T + \bar{V}\frac{T}{K} \cdot dim_E(\Psi, \Pi_{\mathscr{A}}) \cdot \log^2 T.$$

*Proof of Theorem 2.* In this subsection, we present a brief sketch proof for the policy regret bound of BOVL. Our proof strategy consists of the following key steps.

**Step 1: Optimism and error decomposition.** We firstly show that with high probability, the confidence sets $\Psi^{t-1}$ contains the true response $\mu^*$ and $\mathcal{F}^{t-1}(\pi, \nu)$ contains $Q^{\pi,\nu}$ for all $\pi, \nu$ and $t \in [T]$ (Lemma D.1). This naturally comes from martingale concentration and the choice of confidence radii $\alpha$ and $\beta$ in BOVL. Let

$$(f^t, \mu^t) = \underset{f \in \mathcal{F}^{t-1}(\pi^t, \mu(\pi^t)), \mu \in \Psi^{t-1}}{\arg\max} f_1(s_1, \pi^t, \mu(\pi^t)).$$

Notice that switching a policy or playing the same policy but for less than $m$ consecutive episodes adds $\mathcal{O}(\bar{V})$ to the policy regret in the worst case. There are at most $mK$ times BOVL perform such behavior ($K$ policy switches + $(m-1)K$ times playing a policy "immaturally"). Thus we have,

$$PR(T + (m-1)K) - mK\bar{V}$$
$$\leq \sum_{t=1}^{T} V_1^{\pi^*, \mu^*(\pi^*)}(s_1) - V_1^{\pi^t, \mu^*(\pi^t)}(s_1)$$
$$\leq \sum_{t=1}^{T} \max_{\mu \in \Psi^{t-1}} \max_{f \in \mathcal{F}^{t-1}(\pi^*, \mu(\pi^*))} f_1(s_1, \pi^*, \mu(\pi^*)) - V_1^{\pi^t, \mu^*(\pi^t)}(s_1)$$
$$\leq \sum_{t=1}^{T} \max_{\mu \in \Psi^{t-1}} \max_{f \in \mathcal{F}^{t-1}(\pi^t, \mu(\pi^t))} f_1(s_1, \pi^t, \mu(\pi^t)) - V_1^{\pi^t, \mu^*(\pi^t)}(s_1)$$
$$= \sum_{t=1}^{T} \underbrace{f_1^t(s_1, \pi^t, \mu^t(\pi^t)) - V_1^{\pi^t, \mu^*(\pi^t)}(s_1)}_{\xi_1^t},$$

where the second inequality is due to optimism, the third inequality is due to the optimistic planning (Line 4) of Algorithm 1. We have

$$\xi_1^t = \underbrace{f_1^t(s_1, \pi^t, \mu^*(\pi^t)) - V_1^{\pi^t, \mu^*(\pi^t)}(s_1)}_{\zeta_1^t} + \underbrace{f_1^t(s_1, \pi^t, \mu^t(\pi^t)) - f_1^t(s_1, \pi^t, \mu^*(\pi^t))}_{\gamma_1^t}$$

By the standard error decomposition (Lemma B.1), we have

$$\zeta_1^t = \sum_{h=1}^{H} \mathbb{E}_{\pi^t, \mu^*(\pi^t)} \left[ \mathcal{E}_h^{\pi^t, \mu^*(\pi^t)}(f_h^t, f_{h+1}^t)(x_h) \right],$$

where $\mathbb{E}_{\pi, \mu}$ denotes the expectation taken over the random trajectory $(s_1, a_1, b_1, \ldots, s_H, a_H, b_H)$ induced by following $\pi$ and $\mu$, and $x_h = (s_h, a_h, b_h)$. We have

$$\gamma_1^t \leq \bar{V} \cdot \|\mu_1^t(\cdot|s_1, \pi^t) - \mu_1^*(\cdot|s_1, \pi^t)\|_1.$$

Therefore,

$$\mathrm{PR}(T + (m-1)K) - mK\bar{V} \leq \sum_{t=1}^{T} \zeta_1^t + \sum_{t=1}^{T} \gamma_1^t$$

$$\leq \sum_{t=1}^{T} \sum_{h=1}^{H} \mathbb{E}_{\pi^t, \mu^*(\pi^t)} \left[ \mathcal{E}_h^{\pi^t, \mu^*(\pi^t)}(f_h^t, f_{h+1}^t)(x_h) \right] + \bar{V} \sum_{t=1}^{T} \|\mu_1^t(\cdot|s_1, \pi^t) - \mu_1^*(\cdot|s_1, \pi^t)\|_1$$

$$\leq \sum_{t=1}^{T} \sum_{h=1}^{H} \mathcal{E}_h^{\pi^t, \mu^*(\pi^t)}(f_h^t, f_{h+1}^t)(x_h^t) + \tilde{\mathcal{O}}(H\sqrt{T}) + \bar{V} \sum_{t=1}^{T} \|\mu_1^t(\cdot|s_1, \pi^t) - \mu_1^*(\cdot|s_1, \pi^t)\|_1, \qquad (9)$$

where the last inequality follows the standard Azuma-Hoelfding's inequality.

**Step 2: Control the number of batches with large in-batch loss.**  A batch $j$ starts at episode $t_j + 1$ and ends at episode $t_{j+1}$, where $t_j := j\lfloor T/K \rfloor + 1$. Let us denote by $B_j$ the set of all episodes within batch $j$, i.e.,

$$B_j := [t_j + 1, t_{j+1}] := \{t_j + 1, t_j + 2, \ldots, t_{j+1}\}$$

Let us define the "normalized" in-batch losses for all batch $j \in \{0, \ldots, K-1\}$,

$$\Delta_j := \frac{1}{\alpha} \sum_{i=t_j+1}^{t_{j+1}} \mathcal{E}_h^{\pi^i, \mu^i(\pi^i)}(f_h^i, f_{h+1}^i)(x_h^i)^2 \qquad (10)$$

$$\Lambda_j := \frac{1}{\beta} \sum_{i=t_j+1}^{t_{j+1}} \|\mu_1^i(\pi^i) - \mu_1^*(\pi^i)\|_1^2. \qquad (11)$$

Define $\mathcal{K}^s$ be the set of all batches $j \in \{0, \ldots, K-1\}$ with large in-batch bias-centered squared loss, i.e.,

$$\mathcal{K}^s := \{j \in \{0, \ldots, K-1\} : \Delta_j \geq 3\}.$$

Similarly, define $\mathcal{K}^{tv}$ be the set of all batches $j \in \{0, \ldots, K-1\}$ with large in-batch TV distance, i.e.,

$$\mathcal{K}^{tv} := \{j \in \{0, \ldots, K-1\} : \Lambda_j \geq \Omega(1)\}$$

By Lemma F.2 and Lemma F.4, with probability at least $1 - \delta$, we have

$$|\mathcal{K}^s| \lesssim \dim_E(\mathcal{F}, \Pi_{\mathscr{A}}) \cdot \log T \log(T\bar{V}^2). \qquad (12)$$

$$|\mathcal{K}^{tv}| \lesssim \dim_E(\Psi, \Pi_{\mathscr{A}}) \log^2 T. \qquad (13)$$

We also define the complement sets $\bar{\mathcal{K}}^s = \{0, \ldots, K-1\}/\mathcal{K}^s$ and $\bar{\mathcal{K}}^{tv} = \{0, \ldots, K-1\}/\mathcal{K}^{tv}$.

**Step 3: Utilize the design of our confidence sets.**  Recall that, for any batch $j \in [K]$, our construction of the confidence sets at Line 7 forces $f^{t_j+1}$ and $\pi^{t_j+1}$ to have small bias-centered squared loss which is a proxy for squared Bellman error. By standard martingale concentration inequalities (Lemma E.1), we can show that, with probability at least $1 - \delta$,

$$\forall j \in \{0, \ldots, K-1\}, \sum_{i=1}^{t_j} \mathcal{E}_h^{\pi^{t_j+1}, \mu^*(\pi^{t_j+1})}(f_h^{t_j+1}, f_{h+1}^{t_j+1})(x_h^i)^2 \lesssim \alpha.$$

Combining with the construction of $\bar{\mathcal{K}}^s$, we have

$$\forall t \in \bigcup_{j \in \bar{\mathcal{K}}^s} B_j, \sum_{i=1}^{t-1} \mathcal{E}_h^{\pi^t, \mu^*(\pi^t)}(f_h^t, f_{h+1}^t)(x_h^i)^2 \lesssim \alpha. \qquad (14)$$

By Lemma D.3, the construction of the confidence set $\Psi^t$ in Algorithm 1, and Assumption 3.4, we have with probability at least $1 - \delta$,

$$\forall j \in \{0, \ldots, K-1\}, \sum_{i=1}^{t_j} \|\mu_1^{t_j+1}(\cdot|s_1, \pi^i) - \mu_1^*(\cdot|s_1, \pi^i)\|_1^2 \lesssim \beta.$$

Similarly, by the construction of $\bar{\mathcal{K}}^{tv}$, we have

$$\forall t \in \bigcup_{j \in \bar{\mathcal{K}}^{tv}} B_j, \sum_{i=1}^{t-1} \|\mu_1^t(\cdot|s_1, \pi^i) - \mu_1^*(\cdot|s_1, \pi^i)\|_1^2 \lesssim \beta. \tag{15}$$

**Step 4: Establishing the relations between Equation (14), Equation (15) with Equation (9).**  So far, we want to bound Equation (9) while we know Equation (14) and Equation (15). Note that the terms in Equation (9) look similar to the corresponding terms in Equation (14) and Equation (15), except that the terms in Equation (14) and Equation (15) control the losses of the models at time $t$ for all collected data up to time $t-1$. It turns out the Eluder conditions will exactly help us with establishing the relations between Equation (14), Equation (15) with Equation (9). In particular, by Condition 1, Equation (14) implies that

$$\sum_{t \in \bigcup_{j \in \bar{\mathcal{K}}^s} B_j} \mathcal{E}_h^{\pi^t, \mu^*(\pi^t)}(f_h^t, f_{h+1}^t)(x_h^t)^2 \lesssim \dim_E(\mathcal{F}, \Pi_{\mathscr{A}}) \cdot \alpha \log T.$$

Therefore, by Cauchy-Schwartz inequality, we have

$$\sum_{t \in \bigcup_{j \in \bar{\mathcal{K}}^s} B_j} \mathcal{E}_h^{\pi^t, \mu^*(\pi^t)}(f_h^t, f_{h+1}^t)(x_h^t) \lesssim \sqrt{T \cdot \dim_E(\mathcal{F}, \Pi_{\mathscr{A}}) \cdot \alpha \log T}. \tag{16}$$

By a similar chain of arguments, using Condition 2 and Cauchy-Schwartz inequality, Equation (15) implies

$$\sum_{t \in \bigcup_{j \in \bar{\mathcal{K}}^{tv}} B_j} \|\mu_1^t(\cdot|s_1, \pi^t) - \mu_1^*(\cdot|s_1, \pi^t)\|_1 \lesssim \sqrt{T \cdot \dim_E(\Psi, \Pi_{\mathscr{A}}) \cdot \beta \log T}. \tag{17}$$

We are now ready to bound the terms in Equation (9) using Equation (16) and Equation (17). We have

$$\sum_{t=1}^{T} \mathcal{E}_h^{\pi^t, \mu^*(\pi^t)}(f_h^t, f_{h+1}^t)(x_h^t) = \sum_{t \in \bigcup_{j \in \bar{\mathcal{K}}^s} B_j} \mathcal{E}_h^{\pi^t, \mu^*(\pi^t)}(f_h^t, f_{h+1}^t)(x_h^t) + \sum_{t \in \bigcup_{j \in \mathcal{K}^s} B_j} \mathcal{E}_h^{\pi^t, \mu^*(\pi^t)}(f_h^t, f_{h+1}^t)(x_h^t)$$

$$\leq \sqrt{T \cdot \dim_E(\mathcal{F}, \Pi_{\mathscr{A}}) \cdot \alpha \log T} + \bar{V}\frac{T}{K}|\mathcal{K}^s|$$

$$\leq \sqrt{T \cdot \dim_E(\mathcal{F}, \Pi_{\mathscr{A}}) \cdot \alpha \log T} + \bar{V}\frac{T}{K}\dim_E(\mathcal{F}, \Pi_{\mathscr{A}}) \cdot \log T \log(T\bar{V}^2)$$

where the first inequality follows from Equation (16) and the second inequality is due to Equation (12). Similarly, we have

$$\sum_{i=1}^{T} \|\mu_1^t(\cdot|s_1, \pi^t) - \mu_1^*(\cdot|s_1, \pi^t)\|_1 = \sum_{t \in \bigcup_{j \in \bar{\mathcal{K}}^{tv}} B_j} \|\mu_1^t(\cdot|s_1, \pi^t) - \mu_1^*(\cdot|s_1, \pi^t)\|_1 + \sum_{t \in \bigcup_{j \in \mathcal{K}^{tv}} B_j} \|\mu_1^t(\cdot|s_1, \pi^t) - \mu_1^*(\cdot|s_1, \pi^t)\|_1$$

$$\leq \sqrt{T \cdot \dim_E(\Psi, \Pi_{\mathscr{A}}) \cdot \beta \log T} + \frac{T}{K}\dim_E(\Psi, \Pi_{\mathscr{A}}) \log^2 T,$$

where the inequality follows from Equation (17) and Equation (13). Rearranging the terms and rescaling $\delta$ complete our proof.

$$\square$$

## B. Technical Lemmas

We present a standard value-policy error decomposition that straightforwardly generalizes the value-function error decomposition in (Jiang et al., 2017; Jin et al., 2021), from an arbitrary value function and its greedy policy to an arbitrary value function and an *arbitrary* policy.

**Lemma B.1** (Value-policy error decomposition). *For any $f \in \mathcal{F}$ and any policy $\pi, \mu$, we have*

$$f_1(s_1, \pi_1, \mu_1) - V_1^{\pi,\mu}(s_1) = \sum_{h=1}^{H} \mathbb{E}_{\pi,\mu} \left[ (f_h - \mathbb{T}_h^{\pi,\mu} f_{h+1})(s_h, a_h, b_h) \right].$$

*Proof of Lemma B.1.* We have

$$
\begin{aligned}
f_1(s_1, \pi_1, \mu_1) - V_1^{\pi,\mu}(s_1) &= \mathbb{E}_{\pi,\mu} \left[ f_1 - Q_1^{\pi,\mu} \right] \\
&= \mathbb{E}_{\pi,\mu}[f_1] - \mathbb{E}_{\pi,\mu} \left[ \mathbb{T}_1^{\pi,\mu} Q_2^{\pi,\mu} \right] \\
&= \mathbb{E}_{\pi,\mu}[f_1] - \mathbb{E}_{\pi,\mu} \left[ \mathbb{T}_1^{\pi,\mu} f_2 \right] + \mathbb{E}_{\pi,\mu} \left[ \mathbb{T}_1^{\pi,\mu} \{f_2 - Q_2^{\pi,\mu}\} \right] \\
&= \mathbb{E}_{\pi,\mu} \left[ f_1 - \mathbb{T}_1^{\pi,\mu} f_2 \right] + \mathbb{E}_{\pi,\mu} \left[ f_2 - Q_2^{\pi,\mu} \right] \\
&= \dots \\
&= \sum_{h=1}^{H} \mathbb{E}_{\pi,\mu} \left[ f_h - \mathbb{T}_h^{\pi,\mu} f_{h+1} \right],
\end{aligned}
$$

where the first equation follows from the Bellman equation, and the last equation follows from induction. $\square$

The following lemma controls the running sum of zero-mean random variables by their variance, typically known as the Freedman inequality (Freedman, 1975). The proof for Freedman's inequality is elementary, which could be found in e.g. (Nguyen-Tang & Arora, 2024b).

**Lemma B.2** (Freedman's inequality). *Let $X_1, \dots, X_T$ be the sequence of* any *random real-valued variables. Denote $\mathbb{E}_t[\cdot] = \mathbb{E}[\cdot | X_1, \dots, X_{t-1}]$. Assume that $X_t \le R$ for some $R > 0$ and $\mathbb{E}_t[X_t] = 0$ for all $t$. Define the random variables*

$$S := \sum_{t=1}^{T} X_t, \quad V := \sum_{i=1}^{T} \mathbb{E}_t[X_t^2].$$

*Then for any $\delta > 0$, with probability at least $1 - \delta$, for any $\lambda \in [0, 1/R]$,*

$$S \le (e-2)\lambda V + \frac{\ln(1/\delta)}{\lambda}.$$

We define the $\sigma$-algebra $\mathscr{B}_h^k = \sigma \left( \cup_{t \in [k-1], j \in [H]} \mathcal{D}_j^t \cup \{(s_j^k, a_j^k, b_j^k, r_j^k)\}_{j \in [h-1]} \cup (s_h^k, a_h^k, b_h^k) \right)$ denote $\mathbb{E}_{k,h}[\cdot] := \mathbb{E}[\cdot | \mathscr{B}_h^k]$. The following lemma establishes the variance condition on the excess TD loss, a TD analogous to the variance condition that is widely used in the empirical process theory (Massart, 2000).

**Lemma B.3** ((Nguyen-Tang & Arora, 2024b, Lemma B.1)). *For any $\mathscr{B}_h^k$-measurable policy $\pi, \mu$ and any $\mathscr{B}_h^k$-measurable function $f \in \mathcal{F}$, we have*

$$
\begin{aligned}
\mathbb{E}_{k,h}[\Delta l_{\pi,\mu}(f_h, f_{h+1})(z_h^k)] &= \mathcal{E}_h^{\pi,\mu}(f_h, f_{h+1})(x_h^k)^2, \\
\mathbb{E}_{k,h}[\Delta l_{\pi,\mu}(f_h, f_{h+1})(z_h^k)^2] &\le 36\bar{V}^2 \mathcal{E}_h^{\pi,\mu}(f_h, f_{h+1})(x_h^k)^2.
\end{aligned}
$$

---

**Algorithm 2** Meta-algorithm

---
1: **for** $t = 1, 2, \ldots, T$ **do**
2:     Choose $\pi^t$ as a deterministic function of $\{(\pi^i, \tau^i)\}_{i \in [t-1]}$
3:     Execute $\pi^t$ and collect $\tau^t \sim \mathbb{P}_{\theta^*}^{\pi^t}$
4: **end for**

---

## C. Maximum Likelihood Estimation

In this section, we analyze the maximum likelihood estimation (MLE) in the following meta-algorithm in Algorithm 2. Here, a trajectory distribution is uniquely induced by a policy $\pi \in \Pi$ and a parameter $\theta \in \Theta$, denoted by $\mathbb{P}_\theta^\pi$.

We generalize the MLE guarantees from the tabular case in (Liu et al., 2022) to the general model $\mathbb{P}_\theta^\pi$. To do so, we need to define the bracketing numbers of $\Phi$ for our model in a way that makes sense and generalizes the results in (Liu et al., 2022), which we present as follows.

**Definition 6.** $\{[l_i, u_i], i \in [N]\}$ where $l_i, u_i : \Pi \times \mathcal{T} \to \mathbb{R}_+$ is an $\epsilon$-bracketing cover of $\Theta$ if,

- There exists an absolute constant $c > 0$ such that $\max_{i \in [N]} \sup_{\pi \in \Pi} \|u_i(\pi, \cdot)\|_1 \le c$, and

- For any $\theta \in \Theta$, there exists $i \in [N]$ such that $l_i(\pi, \tau) \le \mathbb{P}_\theta^\pi(\tau) \le u_i(\pi, \tau), \forall (\pi, \tau) \in (\Pi, \mathcal{T})$, and

- $\max_{i \in [N]} \sup_\pi \|l_i(\pi, \cdot) - u_i(\pi, \cdot)\|_1 \le \epsilon$.

The smallest such $N$, denoted by $N_\Theta(\epsilon)$, is called the $\epsilon$-bracketing number of $\Theta$ (w.r.t. the model $\mathbb{P}_\theta^\pi$).

**Remark 3.** The upper bracket $u_i(\pi, \cdot)$ in Definition 6 does not need to be a valid probability over $\mathcal{T}$, but its sum over $\mathcal{T}$ must be bounded from above by an absolute constant, which is crucial for our model. This small detail is missing from the proof of Proposition 14 of (Liu et al., 2022) because all upper brackets in the tabular model of (Liu et al., 2022) is a valid probability.

**Remark 4.** When the trajectory distribution $\mathbb{P}_\theta^\pi$ can be factorized as $\mathbb{P}_\theta^\pi(\tau) = f_\theta(\tau) \times \pi(\tau)$, our bracketing number in Definition 6 reduces to the bracketing number defined in (Liu et al., 2023) (the paragraph right after their Definition 2.2). In other words, we do not need this factorization in our definition of bracketing numbers and our results.

Our first result, similar to (Liu et al., 2022, Proposition 13), states that the log-likelihood of the true model is close to the maximum log-likelihood, up to an estimation error that is driven by the complexity of the model class.

**Lemma C.1.** Fix any $\delta \in (0, 1)$ and $T$. With probability at least $1 - \delta$, for all $t \in [T]$ and $\theta \in \Theta$,

$$\sum_{i=1}^t \log \frac{\mathbb{P}_\theta^{\pi^i}(\tau^i)}{\mathbb{P}_{\theta^*}^{\pi^i}(\tau^i)} \lesssim \log\left(N_\Theta(1/T)T/\delta\right).$$

Our second result, similar to (Liu et al., 2022, Proposition 14), specifies how the empirical log-likelihood between a model and the true model controls the total variation distance between the two models.

**Lemma C.2.** Fix any $\delta \in (0, 1)$ and $T$. With probability at least $1 - \delta$, for all $t \in [T]$ and all $\theta \in \Theta$,

$$\sum_{i=1}^t \|\mathbb{P}_\theta^{\pi^i} - \mathbb{P}_{\theta^*}^{\pi^i}\|_1^2 \lesssim \sum_{i=1}^t \log \frac{\mathbb{P}_{\theta^*}^{\pi^i}(\tau^i)}{\mathbb{P}_\theta^{\pi^i}(\tau^i)} + \log\left(N_\Theta(1/T)T/\delta\right).$$

*Proof of Lemma C.1 and Lemma C.2.* By the definition of bracketing numbers in Definition 6, for any $\theta \in \Theta$, there exists $i \in [N_\Theta(1/T)]$ such that

- $\sup_{\pi \in \Pi} \|u_i(\pi, \cdot)\|_1 \le c$ for some absolute constant $c$, and

- $u_i(\pi, \tau) \ge \mathbb{P}_\theta^\pi(\tau), \forall (\pi, \tau) \in \Pi \times \mathcal{T}$, and

- $\sup_{\pi \in \Pi} \|u_i(\pi, \cdot) - \mathbb{P}_\theta^\pi(\cdot)\|_1 \leq 1/T$.

The proof of Lemma C.1 and Lemma C.2 then closely follow that of Proposition 13 and Proposition 14 of (Liu et al., 2022), respectively, by replacing their $f_\theta \times \pi$ by our $\mathbb{P}_\theta^\pi$. $\qquad\square$

## D. Optimism

**Lemma D.1** (Optimism). *Under Assumption 3.2, Assumption 3.3, Assumption 3.1, Assumption 3.4, with probability at least $1 - \delta$,*

$$\forall (\pi, t) \in \Pi \times [T], \mu^* \in \Psi^t \text{ and } Q^{\pi, \mu^*(\pi)} \in \mathcal{F}^t(\pi, \mu^*(\pi)).$$

*Proof of Lemma D.1.* We will prove that, with probability at least $1 - \delta$, $\forall (\pi, t, h) \in \Pi \times [T] \times [H]$, we have

$$\begin{cases} \sum_{i=1}^{t} l_{\pi, \mu^*(\pi)}(Q_h^{\pi, \mu^*(\pi)}, Q_{h+1}^{\pi, \mu^*(\pi)})(z_h^i) - \inf_{f_h \in \mathcal{F}_h} \sum_{i=1}^{t} l_{\pi, \mu^*(\pi)}(f_h, Q_{h+1}^{\pi, \mu^*(\pi)})(z_h^i) \leq \alpha, \\ -\sum_{i=1}^{t} \log \mu_h^*(b_h^i | s_h^i, \pi^i) + \sup_{\mu_h \in \Psi_h} \sum_{i=1}^{t} \log \mu_h(b_h^i | s_h^i, \pi^i) \leq \beta. \end{cases}$$

The first inequality follows from Lemma D.4 and the second inequality follows from Lemma D.2. $\qquad\square$

### D.1. Optimistic MLE

In this section, we establish the MLE guarantees for our adversary model.

**Lemma D.2.** *Fix any $\delta \in (0, 1)$ $h \in [H]$, and $T \in \mathbb{N}$. With probability at least $1 - \delta$, for all $t \in [T]$ and all $\mu_h \in \Psi_h$,*

$$\sum_{i=1}^{t} \log \frac{\mu_h(b_h^i | s_h^i, \pi^i)}{\mu_h^*(b_h^i | s_h^i, \pi^i)} \lesssim \log \left( N_{\Psi_h}(1/T) T / \delta \right),$$

*where $N_{\Psi_h}(\epsilon)$ is the $\epsilon$-bracketing number of $\Psi_h$ as defined in Definition 5.*

*Proof of Lemma D.2.* Let $\tau = (s_1, a_1, b_1, \ldots, s_h, a_h, b_h)$, and define

$$\mathbb{P}_{\mu_h}^\pi(\tau) = \mathbb{P}_1(s_1)\pi_1(a_1|s_1)\mu_1^*(b_1|s_1, \pi)\mathbb{P}_2(s_2|s_1, a_1, b_1) \ldots \mu_h(b_h|s_h)$$

$\qquad\square$

**Lemma D.3.** *Fix any $\delta \in (0, 1)$ $h \in [H]$, and $T \in \mathbb{N}$. With probability at least $1 - \delta$, for all $t \in [T]$ and all $\mu_h \in \Psi_h$,*

$$\sum_{i=1}^{t} \mathbb{E}_{s_h \sim \mathbb{P}_h^{\pi^i}} \|\mu_h(\cdot|s_h, \pi^i) - \mu_h^*(\cdot|s_h, \pi^i)\|_1 \lesssim \sum_{i=1}^{t} \log \frac{\mu_h^*(b_h^i|s_h^i, \pi^i)}{\mu_h(b_h^i|s_h^i, \pi^i)} + \log \left( N_{\Psi_h}(1/T) T / \delta \right),$$

*where $\mathbb{P}_h^\pi(s_h)$ is the probability over $s_h$ by following $\pi$ and $\{\mu_h^*(\cdot|\cdot, \pi)\}_{h \in [H]}$, and $N_{\Psi_h}(\epsilon)$ is the $\epsilon$-bracketing number of $\Psi_h$ as defined in Definition 5.*

*Proof of Lemma D.2 and Lemma D.3.* We apply Lemma C.1 and Lemma C.2, respectively, with

$$\tau = (s_h, b_h), \theta = \mu_h, \Theta = \Psi_h, \mathbb{P}_\theta^\pi(\tau) = \mu_h(b_h|s_h, \pi)\mathbb{P}_h^\pi(s_h).$$

Given the above realization, note that if $\{[l_i, u_i], i \in [N]\}$ is a $\epsilon$-bracketing cover of $\Psi_h$, in the sense defined in Definition 5, then $\{[\tilde{l}_i, \tilde{u}_i], i \in [N]\}$ is an $\epsilon$-bracketing cover of $\Psi_h$ in the sense defined in Definition 6, where $\tilde{l}_i(\pi, \tau) := l_i(\pi, s_h, b_h) \times \mathbb{P}_h^\pi(s_h)$ and $\tilde{u}_i(\pi, \tau) = u_i(\pi, s_h, b_h) \times \mathbb{P}_h^\pi(s_h)$. $\qquad\square$

## D.2. Optimistic Value Function

**Lemma D.4.** *Fix any $\epsilon > 0, \delta \in (0,1)$. Under Assumption 3.2 and Assumption 3.3, with probability at least $1 - \delta$, for all $(t, h, \pi) \in [T] \times [H] \times \Pi$,*

$$\sum_{i=1}^{t} l_{\pi,\mu^*(\pi)}(Q_h^{\pi,\mu^*(\pi)}, Q_{h+1}^{\pi,\mu^*(\pi)})(z_h^i) - \inf_{f_h \in \mathcal{F}_h} \sum_{i=1}^{t} l_{\pi,\mu^*(\pi)}(f_h, Q_{h+1}^{\pi,\mu^*(\pi)})(z_h^i)$$

$$\leq 36(e-2)\bar{V}^2 \log(N_{\mathcal{F}}(\epsilon) N_{\Pi_{\mathscr{A}}}(\epsilon) TH/\delta) + 10\bar{V} Lip_Q (1 + Lip_{Adv}) t\epsilon + 2\bar{V} t\epsilon.$$

*Proof of Lemma D.4.* Fix any $(h, \pi, f_h, f_{h+1}) \in [H] \times \Pi \times \mathcal{F}_h \times \mathcal{F}_{h+1}$. By Lemma B.2 and Lemma B.3, we have that with probability at least $1 - \delta$, for all $\tau \in [0, \frac{1}{13\bar{V}^2}]$,

$$\sum_{i=1}^{t} \mathcal{E}_h^{\pi,\mu^*(\pi)}(f_h, f_{h+1})(x_h^i)^2 - \sum_{i=1}^{t} \Delta l_{\pi,\mu^*(\pi)}(f_h, f_{h+1})(z_h^i)$$

$$\leq 36(e-2)\bar{V}^2 \tau \sum_{i=1}^{t} \mathcal{E}_h^{\pi,\mu^*(\pi)}(f_h, f_{h+1})(x_h^i)^2 + \frac{1}{\tau} \log(1/\delta).$$

By setting $\tau = \frac{1}{36(e-2)\bar{V}^2}$, we have that, with probability at least $1 - \delta$,

$$-\sum_{i=1}^{t} \Delta l_{\pi,\mu^*(\pi)}(f_h, f_{h+1})(z_h^i) \leq 36(e-2)\bar{V}^2 \log(1/\delta).$$

By replacing $f_{h+1}$ in the above inequality by $Q_{h+1}^{\pi,\mu^*(\pi)}$ and using $\mathbb{T}_h^{\pi,\mu^*(\pi)} Q_{h+1}^{\pi,\mu^*(\pi)} = Q_h^{\pi,\mu^*(\pi)}$, we have that, with probability at least $1 - \delta$,

$$\sum_{i=1}^{t} l_{\pi,\mu^*(\pi)}(Q_h^{\pi,\mu^*(\pi)}, Q_{h+1}^{\pi,\mu^*(\pi)})(z_h^i) - \sum_{i=1}^{t} l_{\pi,\mu^*(\pi)}(f_h, Q_{h+1}^{\pi,\mu^*(\pi)})(z_h^i) \leq 36(e-2)\bar{V}^2 \log(1/\delta).$$

We have

$$Q_h^{\pi,\mu^*(\pi)}(x_h) - Q_{h+1}^{\pi,\mu^*(\pi)}(s_{h+1}, \pi, \mu^*(\pi)) - Q_h^{\pi',\mu^*(\pi')}(x_h) + Q_{h+1}^{\pi',\mu^*(\pi')}(s_{h+1}, \pi', \mu^*(\pi'))$$

$$= Q_h^{\pi,\mu^*(\pi)}(x_h) - Q_h^{\pi',\mu^*(\pi)}(x_h)$$

$$+ Q_h^{\pi',\mu^*(\pi)}(x_h) - Q_h^{\pi',\mu^*(\pi')}(x_h)$$

$$- Q_{h+1}^{\pi,\mu^*(\pi)}(s_{h+1}, \pi, \mu^*(\pi)) + Q_{h+1}^{\pi,\mu^*(\pi)}(s_{h+1}, \pi', \mu^*(\pi))$$

$$- Q_{h+1}^{\pi,\mu^*(\pi)}(s_{h+1}, \pi', \mu^*(\pi)) + Q_{h+1}^{\pi,\mu^*(\pi)}(s_{h+1}, \pi', \mu^*(\pi'))$$

$$- Q_{h+1}^{\pi,\mu^*(\pi)}(s_{h+1}, \pi', \mu^*(\pi')) + Q_{h+1}^{\pi',\mu^*(\pi)}(s_{h+1}, \pi', \mu^*(\pi'))$$

$$- Q_{h+1}^{\pi',\mu^*(\pi)}(s_{h+1}, \pi', \mu^*(\pi')) + Q_{h+1}^{\pi',\mu^*(\pi')}(s_{h+1}, \pi', \mu^*(\pi'))$$

$$\leq \mathrm{Lip}_Q \cdot \|\pi - \pi'\|_{\Pi_{\mathscr{A}}} + \mathrm{Lip}_Q \cdot \|\mu^*(\pi) - \mu^*(\pi')\|_{\Pi_{\mathscr{B}}} + \mathrm{Lip}_Q \cdot \|\pi - \pi'\|_{\Pi_{\mathscr{A}}} + \mathrm{Lip}_Q \cdot \|\mu^*(\pi) - \mu^*(\pi')\|_{\Pi_{\mathscr{B}}}$$

$$+ \mathrm{Lip}_Q \cdot \|\pi - \pi'\|_{\Pi_{\mathscr{A}}} + \mathrm{Lip}_Q \cdot \|\mu^*(\pi) - \mu^*(\pi')\|_{\Pi_{\mathscr{B}}}$$

$$\leq 3\mathrm{Lip}_Q (1 + \mathrm{Lip}_{Adv}) \cdot \|\pi - \pi'\|_{\Pi_{\mathscr{A}}}$$

Therefore,

$$l_{\pi,\mu^*(\pi)}(Q_h^{\pi,\mu^*(\pi)}, Q_{h+1}^{\pi,\mu^*(\pi)})(z_h^i) - l_{\pi',\mu^*(\pi')}(Q_h^{\pi',\mu^*(\pi')}, Q_{h+1}^{\pi',\mu^*(\pi')})(z_h^i) \leq 6\bar{V} \mathrm{Lip}_Q (1 + \mathrm{Lip}_{Adv}) \cdot \|\pi - \pi'\|_{\Pi_{\mathscr{A}}}.$$

Similarly, we have

$$l_{\pi,\mu^*(\pi)}(f_h, Q_{h+1}^{\pi,\mu^*(\pi)})(z_h^i) - l_{\pi',\mu^*(\pi')}(f_h', Q_{h+1}^{\pi',\mu^*(\pi')})(z_h^i) \leq 4\bar{V} \mathrm{Lip}_Q (1 + \mathrm{Lip}_{Adv}) \cdot \|\pi - \pi'\|_{\Pi_{\mathscr{A}}} + 2\bar{V} \|f - f'\|_{\infty}.$$

By applying a union bound and a standard discretization, we have, with probability at least $1 - \delta$, for all $(t, h, \pi) \in [T] \times [H] \times \Pi$,

$$\sum_{i=1}^{t} l_{\pi,\mu^*(\pi)}(Q_h^{\pi,\mu^*(\pi)}, Q_{h+1}^{\pi,\mu^*(\pi)})(x_h^i) - \inf_{f_h \in \mathcal{F}_h} \sum_{i=1}^{t} l_{\pi,\mu^*(\pi)}(f_h, Q_{h+1}^{\pi,\mu^*(\pi)})(z_h^i)$$

$$\leq 36(e-2)\bar{V}^2 \log(N_{\mathcal{F}}(\epsilon)N_{\Pi_{\mathscr{A}}}(\epsilon)TH/\delta) + 10\bar{V}\mathrm{Lip}_Q(1 + \mathrm{Lip}_{\mathrm{Adv}})t\epsilon + 2\bar{V}t\epsilon.$$

$\square$

## E. In-distribution Error Control

**Lemma E.1.** *Fix any $\epsilon > 0, \delta \in (0, 1)$. Under Assumption 3.2 and Assumption 3.3 , with probability at least $1 - \delta$, for all $(f, \pi, t, h) \in \mathcal{F} \times \Pi_{\mathscr{A}} \times [T] \times [H]$, we have*

$$(a) \sum_{i=1}^{t} \mathcal{E}_h^{\pi,\mu^*(\pi)}(f_h, f_{h+1})(x_h^i)^2 \leq 2 \sum_{i=1}^{t} \Delta l_{\pi,\mu^*(\pi)}(f_h, f_{h+1})(z_h^i)$$

$$+ 144(e-2)\bar{V}^2 \log(2N_{\mathcal{F}}(\epsilon)N_{\Pi_{\mathscr{A}}}(\epsilon)TH/\delta) + 6t(Lip_Q(1 + Lip_{Adv}) + 2)\bar{V}\epsilon,$$

*and*

$$(b) \sum_{i=1}^{t} \mathcal{E}_h^{\pi,\mu^*(\pi)}(f_h, f_{h+1})(x_h^i)^2 \geq \frac{1}{2} \sum_{i=1}^{t} \Delta l_{\pi,\mu^*(\pi)}(f_h, f_{h+1})(z_h^i)$$

$$- 18(e-2)\bar{V}^2 \log(2N_{\mathcal{F}}(\epsilon)N_{\Pi_{\mathscr{A}}}(\epsilon)TH/\delta) - 3t(Lip_Q(1 + Lip_{Adv}) + 2)\bar{V}\epsilon.$$

*Proof of Lemma E.1.* Fix $\epsilon > 0$. Let $\mathcal{F}_\epsilon, \Pi_\epsilon$ be the $\epsilon$-coverings of $\mathcal{F}$ and $\Pi_{\mathscr{A}}$, respectively. Fix any $(f, \pi, t, h) \in \mathcal{F}_\epsilon \times \Pi_\epsilon \times [T] \times [H]$. By Lemma B.2 and Lemma B.3, we have that with probability at least $1 - \delta$, for all $\tau \in [0, \frac{1}{13\bar{V}^2}]$,

$$\left| \sum_{i=1}^{t} \mathcal{E}_h^{\pi,\mu^*(\pi)}(f_h, f_{h+1})(x_h^i)^2 - \sum_{i=1}^{t} \Delta l_{\pi,\mu^*(\pi)}(f_h, f_{h+1})(z_h^i) \right|$$

$$\leq 36(e-2)\bar{V}^2\tau \sum_{i=1}^{t} \mathcal{E}_h^{\pi,\mu^*(\pi)}(f_h, f_{h+1})(x_h^i)^2 + \frac{1}{\tau} \log(2/\delta)$$

By setting $\tau = \frac{1}{72(e-2)\bar{V}^2}$ and $\tau = \frac{1}{36(e-2)\bar{V}^2}$, respectively, the above inequality implies:

$$\sum_{i=1}^{t} \mathcal{E}_h^{\pi,\mu^*(\pi)}(f_h, f_{h+1})(x_h^i)^2 \leq 2 \sum_{i=1}^{t} \Delta l_{\pi,\mu^*(\pi)}(f_h, f_{h+1})(z_h^i) + 144(e-2)\bar{V}^2 \log(2/\delta),$$

$$\sum_{i=1}^{t} \mathcal{E}_h^{\pi,\mu^*(\pi)}(f_h, f_{h+1})(x_h^i)^2 \geq \frac{1}{2} \sum_{i=1}^{t} \Delta l_{\pi,\mu^*(\pi)}(f_h, f_{h+1})(z_h^i) - 18(e-2)\bar{V}^2 \log(2/\delta).$$

Applying a union bound and rescaling $\delta$, we have that with probability at least $1 - \delta$, it holds for all $(f, \pi, t, h) \in \mathcal{F}_\epsilon \times \Pi_\epsilon \times [T] \times [H]$ that

$$\sum_{i=1}^{t} \mathcal{E}_h^{\pi,\mu^*(\pi)}(f_h, f_{h+1})(x_h^i)^2 \leq 2 \sum_{i=1}^{t} \Delta l_{\pi,\mu^*(\pi)}(f_h, f_{h+1})(z_h^i) + 144(e-2)\bar{V}^2 \log(2N_{\mathcal{F}}(\epsilon)N_{\Pi_{\mathscr{A}}}(\epsilon)TH/\delta), \quad (18)$$

$$\sum_{i=1}^{t} \mathcal{E}_h^{\pi,\mu^*(\pi)}(f_h, f_{h+1})(x_h^i)^2 \geq \frac{1}{2} \sum_{i=1}^{t} \Delta l_{\pi,\mu^*(\pi)}(f_h, f_{h+1})(z_h^i) - 18(e-2)\bar{V}^2 \log(2N_{\mathcal{F}}(\epsilon)N_{\Pi_{\mathscr{A}}}(\epsilon)TH/\delta). \quad (19)$$

We will now bound the discretization errors of $\mathcal{E}_h^{\pi,\mu^*(\pi)}(f_h, f_{h+1})$ and $l_{\pi,\mu^*(\pi)}(f_h, f_{h+1})$ when we discretize $\mathcal{F}$ and $\Pi_{\mathscr{A}}$. Fix any $(h, z) \in [H] \times (\mathscr{S} \times \mathscr{A} \times \mathscr{B})$. For any $\pi, \pi', f, f'$, we have

$$\mathcal{E}_h^{\pi,\mu^*(\pi)}(f_h, f_{h+1}) - \mathcal{E}_h^{\pi',\mu^*(\pi')}(f_h', f_{h+1}')$$
$$= (\mathbb{T}_h^{\pi,\mu^*(\pi)} - \mathbb{T}_h^{\pi',\mu^*(\pi)})f_{h+1}' + (\mathbb{T}_h^{\pi',\mu^*(\pi)} - \mathbb{T}_h^{\pi',\mu^*(\pi')})f_{h+1}' + \mathbb{T}_h^{\pi,\mu^*(\pi)}(f_{h+1} - f_{h+1}') + (f_h - f_h')$$
$$\leq \bar{V} \mathrm{Lip}_Q(1 + \mathrm{Lip}_{\mathrm{Adv}})\|\pi - \pi'\|_{\Pi_{\mathscr{A}}} + 2\|f - f'\|_\infty,$$

where the last inequality follows from the Lipschitzness assumption in Assumption 3.3. By the definition of $\epsilon$-coverings, for any $(\pi, f) \in \Pi_{\mathscr{A}} \times \mathcal{F}$, there exist $(\pi', f') \in \Pi_\epsilon \times \mathcal{F}_\epsilon$ such that $\|f - f'\|_\infty \leq \epsilon$ and $\|\pi - \pi'\|_{1,\infty} \leq \epsilon$. Therefore, we have

$$|\mathcal{E}_h^{\pi,\mu^*(\pi)}(f_h, f_{h+1})^2 - \mathcal{E}_h^{\pi',\mu^*(\pi')}(f_h', f_{h+1}')^2|$$
$$= |\mathcal{E}_h^{\pi,\mu^*(\pi)}(f_h, f_{h+1}) - \mathcal{E}_h^{\pi',\mu^*(\pi')}(f_h', f_{h+1}')| \cdot |\mathcal{E}_h^{\pi,\mu^*(\pi)}(f_h, f_{h+1}) + \mathcal{E}_h^{\pi',\mu^*(\pi')}(f_h', f_{h+1}')|$$
$$\leq 2(\bar{V} \mathrm{Lip}_Q(1 + \mathrm{Lip}_{\mathrm{Adv}}) + 2)\bar{V}\epsilon. \tag{20}$$

Similarly, we have

$$|\Delta_{\pi,\mu^*(\pi)}(f_h, f_{h+1}) - \Delta_{\pi',\mu^*(\pi')}(f_h', f_{h+1}')| \leq 2(\bar{V} \mathrm{Lip}_Q(1 + \mathrm{Lip}_{\mathrm{Adv}}) + 2)\bar{V}\epsilon. \tag{21}$$

Combining both Equation (20) and Equation (21) into Equation (18) and Equation (19) completes our proof. $\square$

## F. Handling Batches

Let

$$t_j := j\lfloor T/K \rfloor + 1.$$

Let us first fix any $h \in [H]$. For any $j \in \{0, 1, \ldots, K\}$, we know that

$$\forall t \in B_j := [t_j + 1, t_{j+1}], \pi^t = \pi^{t_j+1}, f^t = f^{t_j+1}, \mu^t = \mu^{t_j+1}, \tag{22}$$

### F.1. For Value Function Approximation

We define the "normalized" in-batch cumulative Bellman errors for each batch $j \in [K]$ as

$$\Delta_j := \frac{1}{\alpha} \sum_{i=t_j+1}^{t_{j+1}} \mathcal{E}_h^{\pi^i,\mu^i(\pi^i)}(f_h^i, f_{h+1}^i)(z_h^i)^2.$$

**Lemma F.1.** *Fix any $C \geq 3$. Under Assumption 3.1 and Condition 1, with probability at least $1 - \delta$,*

$$|\{j \in [K] : \frac{C}{2} \leq \Delta_j \leq C\}| \lesssim dim_E(\mathcal{F}, \Pi_{\mathscr{A}}) \cdot \log T.$$

*Proof of Lemma F.1.* Let $j_1 \leq \ldots j_M$ be all the batches $j$ such that $\frac{C}{2} \leq \Delta_j \leq C$. Thus, we have

$$\sum_{i \in \bigcup_{m \in [M]} B_{j_m}} \mathcal{E}_h^{\pi^i,\mu^i(\pi^i)}(f_h^i, f_{h+1}^i)(z_h^i)^2 \geq \frac{MC\alpha}{2}. \tag{23}$$

By Lemma E.1, with probability at least $1 - \delta$,

$$\forall (j, h) \in [K] \times [H], \sum_{i=1}^{t_j} \mathcal{E}_h^{\pi^{t_j+1},\mu^*(\pi^{t_j+1})}(f_h^{t_j+1}, f_{h+1}^{t_j+1})(x_h^i)^2 \leq 2 \sum_{i=1}^{t_j} \Delta l_{\pi^{t_j+1},\mu^*(\pi^{t_j+1})}(z_h^i) + \alpha \leq 3\alpha,$$

where the last inequality due to Algorithm 1 that the version space $\mathcal{C}^t$ is updated at episode $t_j$, and thus $\sum_{i=1}^{t_j} \Delta l_{\pi^{t_j+1},\mu^*(\pi^{t_j+1})}(z_h^i) \leq \alpha$. Under the same event as the above inequality, for all $m \in [M]$ and all $t \in B_{j_m} = [j_m+1, j_{m+1}]$, we have

$$
\sum_{i \in [t-1] \cap \bigcup_{m \in [M]} B_{j_m}} \mathcal{E}_h^{\pi^t,\mu^*(\pi^t)}(f_h^t, f_{h+1}^t)(x_h^i)^2 \leq \sum_{i \in [t-1]} \mathcal{E}_h^{\pi^t,\mu^*(\pi^t)}(f_h^t, f_{h+1}^t)(x_h^i)^2
$$

$$
= \sum_{i=1}^{t_{j_m}} \mathcal{E}_h^{\pi^t,\mu^*(\pi^t)}(f_h^t, f_{h+1}^t)(x_h^i)^2 + \sum_{i=t_{j_m}+1}^{t-1} \mathcal{E}_h^{\pi^t,\mu^*(\pi^t)}(f_h^t, f_{h+1}^t)(x_h^i)^2
$$

$$
= \sum_{i=1}^{t_{j_m}} \mathcal{E}_h^{\pi^{t_{j_m}+1},\mu^*(\pi^{t_{j_m}+1})}(f_h^{t_{j_m}+1}, f_{h+1}^{t_{j_m}+1})(x_h^i)^2 + \sum_{i=t_{j_m}+1}^{t-1} \mathcal{E}_h^{\pi^t,\mu^*(\pi^t)}(f_h^t, f_{h+1}^t)(x_h^i)^2
$$

$$
\leq 3\alpha + C\alpha = (C+3)\alpha.
$$

By Condition 1, the above inequalities imply that

$$
\sum_{i \in \bigcup_{m \in [M]} B_{j_m}} \mathcal{E}_h^{\pi^i,\mu^*(\pi^i)}(f_h^i, f_{h+1}^i)(x_h^i)^2 \lesssim d(\mathcal{F}, \Pi) \cdot (C+3)\alpha \log T
$$

Combining the above inequality with Equation (23) gives

$$
M \lesssim \dim_E(\mathcal{F}, \Pi_{\mathscr{A}}) \cdot \log T,
$$

for $C \geq 3$. $\qquad\square$

**Lemma F.2.** *Under Assumption 3.1 and Condition 1, with probability at least $1 - \delta \log(T\bar{V}^2)$,*

$$
|\{j \in [B] : \Delta_j \geq 3\}| \lesssim \dim_E(\mathcal{F}, \Pi_{\mathscr{A}}) \cdot \log T \log(T\bar{V}^2).
$$

*Proof.* Note that, $\Delta_j \leq \frac{(T/K)\bar{V}^2}{\alpha} \lesssim T\bar{V}^2$. In addition, $[3, B\bar{V}^2] \subseteq \bigcup_{i=1}^{\log(T\bar{V}^2/3)}[3 \times 2^{i-1}, 3 \times 2^i)$. Applying Lemma F.1 to each $[3 \times 2^{i-1}, 3 \times 2^i)$ and using a union bound completes our proof. $\qquad\square$

### F.2. For Adversary Approximation

We define the normalized in-batch cumulative TV distance for each batch $j$ as

$$
\Lambda_j := \frac{1}{\beta} \sum_{i=t_j+1}^{t_{j+1}} \|\mu_1^i(\pi^i) - \mu_1^*(\pi^i)\|_1^2, \forall j \in [K].
$$

**Lemma F.3.** *For any $C$ that is greater than some absolute constant $C_0$, under Assumption 3.4 and Condition 2, with probability at least $1 - \delta$,*

$$
\left|\left\{j \in [K] : \frac{C}{2} \leq \Delta_j \leq C\right\}\right| \lesssim \dim_E(\Psi, \Pi_{\mathscr{A}}) \cdot \log T.
$$

*Proof of Lemma F.3.* By Lemma D.3, there is an absolute constant $c$ such that with probability at least $1 - \delta$, for all $j \in [B]$, we have

$$
\sum_{i=1}^{t_j} \|\mu_1^{t_j+1}(\pi^i) - \mu_1^*(\pi^i)\|_1^2 \lesssim \sum_{i=1}^{t_j} \log \frac{\mu_1^*(b_1^i|s_1^i, \pi^i)}{\mu_1^{t_j+1}(b_1^i|s_1^i, \pi^i)} + \beta \leq \sup_{\mu \in \Psi} \sum_{i=1}^{t_j} \log \frac{\mu_1(b_1^i|s_1^i, \pi^i)}{\mu_1^{t_j+1}(b_1^i|s_1^i, \pi^i)} + \beta \leq 2\beta, \quad (24)
$$

where the second inequality follows from Assumption 3.4 and the last inequality is due to the control of the log-likelihood loss at the beginning $t_j + 1$ of batch $j$ in Algorithm 1.

Fix $C > 0$. Let $j_1 < j_2 < \ldots < j_M$ be all the batches $j$ such that

$$\frac{C\beta}{2} \leq \sum_{i \in B_j} \|\mu_1^i(\pi^i) - \mu_1^*(\pi^i)\|_1^2 \leq C\beta. \tag{25}$$

Thus, we have

$$\sum_{i \in \bigcup_{m \in [M]} B_{j_m}} \|\mu_1^i(\pi^i) - \mu_1^*(\pi^i)\|_1^2 \geq \frac{MC\beta}{2}. \tag{26}$$

Under the same event for Equation (24), for all $t \in B_{j_m} = [j_m + 1, j_{m+1}]$, we have

$$\sum_{i \in [t-1] \cap \bigcup_{m \in [M]} B_{j_m}} \|\mu_1^t(\pi^i) - \mu_1^*(\pi^i)\|_1^2 \leq \sum_{i=1}^{t-1} \|\mu_1^t(\pi^i) - \mu_1^*(\pi^i)\|_1^2$$

$$= \sum_{i=1}^{j_m} \|\mu_1^t(\pi^i) - \mu_1^*(\pi^i)\|_1^2 + \sum_{i=j_m+1}^{t-1} \|\mu_1^t(\pi^i) - \mu_1^*(\pi^i)\|_1^2$$

$$= \sum_{i=1}^{j_m} \|\mu_1^{j_m+1}(\pi^i) - \mu_1^*(\pi^i)\|_1^2 + \sum_{i=j_m+1}^{t-1} \|\mu_1^t(\pi^i) - \mu_1^*(\pi^i)\|_1^2$$

$$\leq \mathcal{O}(1)\beta + C\beta.$$

where the second equality comes from $\mu^t = \mu^{j_m+1}, \forall t \in B_{j_m}$, the second inequality comes from Equation (24), and Equation (25).

Therefore, by Condition 2,

$$\sum_{i \in \bigcup_{m \in [M]} B_{j_m}} \|\mu_1^i(\pi^i) - \mu_1^*(\pi^i)\|_1^2 \leq \dim_E(\Psi, \Pi_{\mathscr{A}})(\mathcal{O}(1) + C)\beta \log(T).$$

Combining with Equation (26), we have

$$M \lesssim \dim_E(\Psi, \Pi_{\mathscr{A}}) \log T$$

as long as $C$ is greater than some absolute constant. $\square$

**Lemma F.4.** *There is an absolute constant $C_0$ such that, with probability at least $1 - \delta \log(T)$,*

$$|\{j \in [K] : \Lambda_j \geq C_0\}| \lesssim dim_E(\Psi, \Pi_{\mathscr{A}}) \log^2 T.$$

*Proof of Lemma F.4.* Note that $\Delta_j \leq \frac{K}{\beta} \leq K \leq T$, where $\beta$ is $\Omega(1)$. We have $[C_0, T] \subset \bigcup_{i=0}^{\log(T/C_0)} [C_0 2^{i-1}, C_0 2^i)$. Applying Lemma F.3 to each interval $[C_0 2^{i-1}, C_0 2^i)$ and combining them via a union bound completes the proof. $\square$

## G. Proofs of Supporting Lemmas in the Main Paper

*Proof of Lemma 3.1.* Bounding the log $\epsilon$-covering of a linear model is standard, e.g., see (Wainwright, 2019). Since $\mathcal{F}$ is the product space of $H$ function classes, the log $\epsilon$-covering number of $\mathcal{F}$ can be bounded by the sum of the bounds on the log $\epsilon$-covering number of all $\mathcal{F}_h$. $\square$

*Proof of Lemma 3.2.* Note that by the structure of $\Psi$, each $\mu_h \in \Psi_h$, $\mu_h(\cdot|s,\pi)$ depends on $(\pi, h, s)$ only via the vector $w_{sh}^{\pi}$, where $\Phi$ is completely independent of $(\pi, h, s)$. Thus, the bracketing number of $\Psi$ is bounded by the bracketing number of $\{\Phi \in \mathbb{R}_+^{B \times d_{adv}}, \|\Phi\|_{\infty} = \mathcal{O}(1)\}$. We can construct an $\epsilon$-bracketing cover of that set from an $\epsilon$-cover w.r.t. $\|\cdot\|_{\infty}$. In particular, for each $\Phi \in \mathbb{R}^{B \times d_{adv}}$, we write $\Phi = (\Phi_1, \ldots \Phi_B)$ where $\Phi_j \in \mathbb{R}^{d_{adv}}$. Each coordinate $\Phi_j$ can be $\epsilon$-covered by at most $\mathcal{O}\left(\left(\frac{1}{\epsilon}\right)^{d_{adv}}\right)$ balls of radius $\epsilon$ (Wainwright, 2019). From each $\epsilon$-cover, we can construct one $\epsilon$-bracket. Indeed, let say $\tilde{\Phi}_j$ is an $\epsilon$-cover of $\Phi_j$, i.e., $\|\tilde{\Phi}_j - \Phi_j\|_{\infty} \leq \epsilon$. Then, $[\tilde{\Phi}_j - \epsilon\mathbb{1}, \tilde{\Phi}_j + \epsilon\mathbb{1}]$ is a valid $\epsilon$-bracket of $\Phi_j$, where $\mathbb{1}$ denotes the $d_{adv}$-dimensional vector with all coordinates of 1. Then, perform a combinatorial count over all $j \in [B]$ and a careful rescaling of $\epsilon$ complete our proof. $\qquad\square$

*Proof of Lemma 5.1 and Lemma 5.2.* With simple algebraic manipulations, the problems in the above lemmas reduce to the standard elliptical potential lemma (Abbasi-Yadkori et al., 2011; Jin et al., 2023). $\qquad\square$

