# OpenReview forum: "Policy-Regret Minimization in Markov Games with Function Approximation"
_ICML.cc/2025/Conference — ICML 2025 poster_

### Official Review · Reviewer_vNWW · 2025-03-10

**Overall Recommendation:** 3

**Summary:**

**Edit post-rebuttal: I thank the authors for their feedback, which answered my questions. I maintain my overall positive score.**

The submission considers Markov games, that is, MDPs where transitions depend on the pair of actions output by two players (the learner and the opponent). It studies a notion of regret called policy regret, which corresponds to comparing what the learner obtains (as measured by the sum of value functions corresponding to the policies output by the learner and the opponent over time) to what the learner would have obtained by playing the same policy $\pi$ over time (as measured by the sum of value functions corresponding to $\pi$ and to the policies output by the opponent facing the constant sequence of policies $\pi$). This notion of regret is the correct counterfactual measure in games and was studied in online learning by Arora et al. (2012), in games by Arora et al. (2018), and extended to Markov games by Nguyen-Tang and Arora (2024). However, the algorithm and regret bounds introduced by the latter reference are for the case of small-scale problems (the sets of states and actions should not be large). The point of the present submission is to provide an algorithm and regret bounds for the case of large-scale problems. This is achieved via considering various notions of complexity (Lipschitzness of value functions and eluder coefficients), that are meaningful especially in the case of linear approximations (for the value functions and for the opponent's strategies). The same restrictions as in Nguyen-Tang and Arora (2024) remain: the opponent should have a bounded memory and be stationary in some sense.

**Claims And Evidence:**

Disclaimer: I can see why (based on some keywords also contained in my own publications) the system assigned this submission to me, but I must declare that I was totally unaware of the specific and advanced line of research featured in this submission before reading it. All what follows is therefore an educated guess only.

All evidence (all proofs) is (are) actually provided in the appendices. That being said, the flow of the main body (how concepts, definitions, assumptions, examples, etc., are introduced and follow each other) looks coherent.

**Essential References Not Discussed:**

The submission seems to do a good job in discussing earlier works; somehow, this is relatively simple as it forms a direct follow-up to the recent work by Nguyen-Tang and Arora (2024).

**Experimental Designs Or Analyses:**

N/A

**Methods And Evaluation Criteria:**

The setting and the main evaluation criterion (the policy regret) were introduced by Nguyen-Tang and Arora (2024), and make sense in the history of articles about policy regret (Arora et al., 2012 and 2018, mainly). The consideration of linear approximations, which justify most of the assumptions and concepts introduced in this submission, is generally standard in machine learning, and was already used in diverse forms also for MDPs (perhaps not exactly in the way this submission does, but the approach is standard).

**Other Comments Or Suggestions:**

- Section 1.1: if the same intuitions for batching as in Arora et al. (2012) apply, you should acknowledge them here
- Eq. (1): is the $\ell_1$ in the right-hand side norm the total-variation distance?
- Line 161, Bellman operator: I think it takes a function as input, and not an element of $\mathbb{R}^{S \times A \times B}$
- Line 163: what is $z$? I guess a triplet $(s,a,b)$? Then, given the notation introduced right after, it should rather be denoted by $x$
- Line 163: rather ${P}_{h+1}$ (not a P with a double bar)
- Lines 179-180 (first column): I guess you refer to Theorem 3 of Nguyen-Tang and Arora (2024)? Their result is about a $\sqrt{T}$ rate, but with a large constant, which to me is a sublinear policy regret
- Line 266 (both columns): I think it should be $\Psi = \Psi_1 \times \ldots \times \Psi_H$
- Page 5, second column (Definition 5 and Example 3.2): this is actually the point in the submission where I thought that notation was really piling up

**Other Strengths And Weaknesses:**

The main limitation of this work is acknowledged by the authors in an upfront and honest manner in Remark 2: the lack of computational efficiency of the strategy introduced. The approach introduced is therefore only worth for the theory but does not seem to solve real cases of large-scale problems.

I found the exposition generally nice. In particular, it is useful to summarize the results in Table 1, as they are formally stated only in the second column of page 8 (!), due to all notions, concept, and notation to be introduced.

It would be great to better emphasize where exactly the improvements were made possible. To me, the key point would be Definition 4 vs. Assumption 3, is that correct? Also, the use of previous techniques (batching as in Arora et al., 2012, and likelihood fitting, as in Nguyen-Tang and Arora, 2024) could be better acknowledged in the comments on Algorithm 1 on page 6.

The independence of the bound of Theorem 1 in the cardinality of the state and action spaces is a bit of an overstatement: indeed, there are implicit dependencies through the complexity measures introduced. This claim should probably be toned down.

**Questions For Authors:**

I have no specific question given my limited knowledge but I would be grateful to the authors if they could answer or react to issues or questions that I raise above.

**Edit post-rebuttal: I thank the authors for their feedback, which answered my questions. I maintain my overall positive score.**

**Relation To Broader Scientific Literature:**

N/A

**Theoretical Claims:**

All proofs are contained in the appendix, which I did not have time to review in detail (especially given the time I spent on the main body, due to my unfamiliarity with the setting). I think that the sketch of the proof of Theorem 1, that can be read in Appendix A, should have been provided in the main body. How close / related is the batching approach followed here, and formally used to get (2), to the one in Arora et al. (2012)? Somehow, I have the feeling that a part of the intuition remains: exploiting the $m$-bounded memory assumption to get back to standard external regret (for Markov games) on sufficiently long batches, of sizes $K$ proportional to $\sqrt{T}$.

---

> ### Author Rebuttal · Authors · 2025-03-31
>
> Thank you for the positive feedback and the detailed comments.
>
> ---
> > How close / related is the batching approach followed here, and formally used to get (2), to the one in Arora et al. (2012)? Somehow, I have the feeling that a part of the intuition remains: exploiting the m-bounded memory assumption to get back to standard external regret (for Markov games) on sufficiently long batches, of sizes $K$ proportional to $\sqrt{T}$
>
> Yes, the batching is to exploit the m-bounded memory as in Arora et al 2012. The key technical challenge is, however, how to bound the regret within a batch in our case, while  Arora et al 2012 use the worst-cast bound within a batch (and thus obtain T^{⅔} regret).
>
> ---
> > It would be great to better emphasize where exactly the improvements were made possible. To me, the key point would be Definition 4 vs. Assumption 3, is that correct?
>
> The conditions that enable the improvements are Definition 4 vs Assumption 3 (as you mentioned), and the Eluder conditions in Section 5.1, and bracketing number in Definition 5.
>
> ---
> > Also, the use of previous techniques (batching as in Arora et al., 2012, and likelihood fitting, as in Nguyen-Tang and Arora, 2024) could be better acknowledged in the comments on Algorithm 1 on page 6.
>
> Thank you. We will be more clear about it in the revision.
>
> ---
> > The independence of the bound of Theorem 1 in the cardinality of the state and action spaces is a bit of an overstatement: indeed, there are implicit dependencies through the complexity measures introduced. This claim should probably be toned down
>
> Thank you. We will be more clear about it in the revision.
>
> ---
> > Section 1.1: if the same intuitions for batching as in Arora et al. (2012) apply, you should acknowledge them here
>
> We will acknowledge Arora et. al. (2021) there when we mention batching. It’s worth noticing that Arora et. al. (2021) use the worst-case bound on the in-batch data error while bounding the in-batch data error is a key technical challenge in our setting, thus Section 1.1.
>
> ---
> > Eq. (1): is the $\ell_1$ in the right-hand side norm the total-variation distance?
>
> Yes.
>
> ---
> > Line 161, Bellman operator: I think it takes a function as input, and not an element of $\mathbb{R}^{S \times A \times B}$
>
>
> An element of $\mathbb{R}^{S \times A \times B}$ is a function from $S \times A \times B$ to $\mathbb{R}$
>
> ---
>
> > Line 163: what is $z$ I guess a triplet $(s,a,b)$. Then, given the notation introduced right after, it should rather be denoted by $x$.
>
> Yes. It's $x$. Thank you.
>
> ---
>
> > Line 163: rather P_{h+1} (not a P with a double bar)
>
> Thank you.
>
> ---
>
> > Lines 179-180 (first column): I guess you refer to Theorem 3 of Nguyen-Tang and Arora (2024)? Their result is about a
>  rate $\sqrt{T}$, but with a large constant, which to me is a sublinear policy regret
>
> Yes, you are correct, we meant sample-efficient, rather than sublinear policy regret. We’ve revised it to: “policy regret minimization is not sample-efficient against”
>
> ---
> > Line 266 (both columns): I think it should be $\Psi_1 \times \ldots \Psi_H$
>
> Yes. We've revised it accordingly.
>
>
>
>
> ---

---

### Official Review · Reviewer_Vowp · 2025-03-13

**Overall Recommendation:** 4

**Summary:**

The paper introduces the first algorithmic framework for policy regret minimization in Markov games with general function approximation, achieving an $O\sqrt{T}$ policy regret bound for a wide range of problems. This framework extends to both large-scale environments with Eluder-type conditions and tabular cases, where it provides a significantly tighter bound. Additionally, it offers a simple and effective approach for handling reactive adversaries, demonstrating how opponent learning can lead to optimal regret rates in dynamic environments.

**Claims And Evidence:**

Yes, they provide a clear table compared with prior work and support those claim and setting with comprehensive theories.

**Essential References Not Discussed:**

I would like to confirm if adaptive adversary this paper is studying is just the standard adversary in robust RL via adversarial training, where players are modeled as max-min game. If so, then it will be proper that authors should at least including the following related work, where [1] is the fundermental RARL setting, [2] extends the two-player game to Stackelberg game, and the latest work [3] improves 2-player game in robustness.

[1] Lerrel Pinto et al. "Robust Adversarial Reinforcement Learning", ICML, 2017

[2] Peide Huang et al. "Robust Reinforcement Learning as a Stackelberg Game via Adaptively Regularized Adversarial Training", IJCAI, 2022

[3] Juncheng Dung et al. "Variational Adversarial Training Towards Policies with Improved Robustness", AISTATS, 2025

**Experimental Designs Or Analyses:**

Without experimental section, which I think it is good to have, but not required for this paper.

**Methods And Evaluation Criteria:**

This is a pure theoretical paper without any experiments. Although I really appreciate the contributions from theoretical perspectives to this field. One of the improvement of this work is to extend from tabular setting to function approximation. Therefore, I may expect that there will be a simple simulation experiment, even in toy example, to see the alignment between theories and experiments.

**Other Comments Or Suggestions:**

N/A

**Other Strengths And Weaknesses:**

Strengths

* well-written
* extension to function approximation while resulting in tighter bound even in tabular setting

Weakness

**Questions For Authors:**

* In your introduction, authors mention the prior work that establishes the fundamental barriers for policy regret minimization in Markov games. And then highlighting two motivations of this work: (1) the consistent behavior is restricted (2) large state/action space. Could you elaborate more about (1) compared with your method, explaining the mathematical equations.

* Is not "the adversary behavior does not change over time" mentioned in your manuscript a strong assumption in practice? Could you explain more how possible you can get rid of this assumption and whether it is common to limit adversary behavior?

**Relation To Broader Scientific Literature:**

This paper mainly compare with (Nguyen-Tang & Arora, 2024a) with the improvement from varying perspectives, which can be viewed as a more general setting in this direction.

**Theoretical Claims:**

I think the theoretical claims looks smooth and with detailed step by step in the proof.

---

> ### Author Rebuttal · Authors · 2025-03-31
>
> Thank you for your constructive feedback.
>
> ---
>
> > I would like to confirm if adaptive adversary this paper is studying is just the standard adversary in robust RL via adversarial training, where players are modeled as max-min game.
>
> No, the adaptive adversary in our paper adapts to the learner’s past and current strategies, while the adversary in robust RL only adapts to the learner’s current strategy ($m=1$ in our terminology). The settings are also different because we look at policy regret which makes sense in such an adaptive adversary setting. We give a regret bound, not just convergence to equilibrium.
>
> ---
> > In your introduction, authors mention the prior work that establishes the fundamental barriers for policy regret minimization in Markov games. And then highlighting two motivations of this work: (1) the consistent behavior is restricted (2) large state/action space. Could you elaborate more about (1) compared with your method, explaining the mathematical equations.
>
> For any two sequences of the learner’s policies, if the two sequences agree in a certain step $h$ and state $s$, then the adversary’s responses to the two sequences also agree in step $h$ and state $s$. This enables us to decompose the response into states and steps independently thus enabling sample-efficient learning in the tabular case. The idea of this assumption is to encode that the adversary responds similarly to two similar sequences of policies. But it does not work for large state space problems and is too restrictive. The Lipschitzness assumption is much more relaxed. See line 225-261 for our explanation.
>
> ---
>
> > Is not "the adversary behavior does not change over time" mentioned in your manuscript a strong assumption in practice? Could you explain more how possible you can get rid of this assumption and whether it is common to limit adversary behavior?
>
> If we get rid of that assumption, policy regret minimization is not sample-efficient anymore, as proven in Nguyen-Tang & Arora 2024. That’s why we consider it in the current paper. Even though the response function does not change over time, it is still powerful as it can remember the learner’s past and current strategies. The adversary is also given unlimited computation power to come up with whatever response function that they can compute using the learner’s past strategies. The simplest example is general-sum games, where given a policy for the learner, the adversary can use any computational power to compute the best-response policy that maximizes its utility given the learner’s policy.

---

### Official Review · Reviewer_orgF · 2025-03-14

**Overall Recommendation:** 3

**Summary:**

This work explores policy regret minimization and proposes a new algorithm (BOVL) that is more general than past literature in that it deals with a class larger than tabular data Markov games. They use function approximation classes (characterized by Eluder type conditions) which can deal with larger state/action spaces while still providing a bound on policy regret that doesn’t depend on the number of states and actions.

**Claims And Evidence:**

The paper claims the following:
- Building on theoretical guarantees for function approximations, they achieve a regret that is tighter than algorithms for tabular cases introduced in past literature

Evidence:
- They provide proof and theoretical analysis of the regret bound

**Essential References Not Discussed:**

/

**Experimental Designs Or Analyses:**

The paper doesn’t contain experiments.

**Methods And Evaluation Criteria:**

The paper doesn’t include empirical experiments/ evaluations on benchmarks.

**Other Comments Or Suggestions:**

- Line 152: I believe (s) in max operator is not meant to be from A but from S
- Line 315: model mu -> models mu
- Line 361: should phi be of subscript i instead of t? The same for mu in line 423
- Line 369: should it be log T?

**Other Strengths And Weaknesses:**

Strengths:

- Theoretical analysis of the presented algorithm which is lacked in MARL literature
-The regret bounds that are tighter than other algorithms for tabular cases and also the ability to deal with larger state space and action space problems making it applicable to wider range of Markov games
- The paper explained theoretical aspects in a simple way and the theorems, lemmas,..etc were easy to follow and well referenced.

Weaknesses:

-The whole paper is built on Eluder conditions (Condition 1 and 2) which I am not sure how applicable they are (See questions below), if not then the regret bound doesn’t hold
- Remark 2 on the tri-level optimization problem makes the algorithm not tractable to be used in practice
- The paper has lots of tiny mistakes in notations / typos that may hinder understanding

**Questions For Authors:**

1. What is the applicability of the eluder-condition on function approximations in real applications?
2. The paper only mentions linear function approximation, what about the non-linear cases? Does this algorithm scale or not and what are the challenges for extending it?
3. I didn’t understand why the adversary derives its policy based on a sequence of learner policies and not just the current one? I understand that this might be a subset of your setting when m=1 but why in general?
4. I also didn’t understand the importance of the “warm up” step on line 6 of the algorithm

**Relation To Broader Scientific Literature:**

The work is an extension to (Nguyen-Tang & Arora, 2024a). Comparisons are carried out against it which is reasonable, but how does this work compares to other literature in Markov games and adversarial-opponent learning is not addressed.

**Theoretical Claims:**

The theoretical claims seem reasonable to me but I listed some questions below. I also did not verify the proofs.

---

> ### Author Rebuttal · Authors · 2025-03-31
>
> Thank you for your feedback.
>
> ---
> > Comparisons are carried out against it which is reasonable, but how does this work compares to other literature in Markov games and adversarial-opponent learning is not addressed.
>
> Much of the prior work in Markov games and adversarial-opponent learning focus on regret, while we focus on policy regret. The study of policy regret against adaptive adversaries has been settled in prior work, e.g. extensive body of work building on the work of Arora et al. 2012; see the second paragraph in our introduction section. We also refer the reader to the related work section of  Nguyen-Tang & Arora, 2024a for a broader context. Nonetheless, we can expand the discussion of other related work if the reviewer has specific suggestions.
>
> ---
>
> > Remark 2 on the tri-level optimization problem makes the algorithm not tractable to be used in practice
>
> Actually, the optimization can be viewed as a bi-level optimization since $\pi$ and $\mu$ can be combined into one optimization variable. Yes, it is intractable in general, but that is the case for any RL problem with general function approximation. However, the optimization problem is tractable in the linear case.
>
> ---
> > Line 152: I believe (s) in max operator is not meant to be from A but from S
>
> Yes. Thank you for pointing out.
>
> ---
> > Line 361: should phi be of subscript i instead of t? The same for mu in line 423
>
> No. The preconditions in the Eluder conditions evaluate the current models (associated with subscript $t$) in the past data (associated with subscripts $i < t$).
>
> ---
> > Line 369: should it be log T?
>
> No. It's $\log t$. But bounding with $\log T$ is also fine.
>
> ---
> > What is the applicability of the eluder-condition on function approximations in real applications?
>
> Any learning theoretic result needs to define a notion of complexity of the hypothesis class or the task we are learning to give sufficient (and sometimes necessary) conditions for learning. The Eluder dimension is one such measure for online learning and RL problems. The Eluder condition here is used to control (and thus inform) the convergence rate of an optimistic algorithm when function approximation is used. In real applications, if you use optimistic algorithms with a form of function approximation, estimating the eluder condition will provide insight into how quickly your algorithm will converge to an optimal policy.
>
> ---
> > The paper only mentions linear function approximation, what about the non-linear cases? Does this algorithm scale or not and what are the challenges for extending it?
>
> We mention the linear case as a concrete example. The results in the paper apply to any nonlinear case, as long as the Eluder conditions hold. For computational efficiency, our algorithm is efficient for tabular and linear cases, but it is likely not tractable for general function approximation, as is the case for RL with general function approximation.
>
> ---
> > I didn’t understand why the adversary derives its policy based on a sequence of learner policies and not just the current one? I understand that this might be a subset of your setting when m=1 but why in general?
>
> In real-world applications, it is almost always the case that the adversary chooses its policy based on a sequence of the learner's policies. In fact that is a general setting in any multi-step game. For example, consider a spam filter that is trained using online learning, updating its model daily based on newly labeled examples. Spammers act as adaptive adversaries: they probe the system by sending test emails and observing which ones evade detection, then evolve their evasion strategies based on the observed sequence of model updates. Importantly, the adversary’s strategy is not fixed—it adapts over time in response to the entire history of the learner’s policies. This interaction cannot be fully captured by assuming a static or memoryless adversary, since the adversary’s behavior depends on trends or patterns in the learner’s updates. Other similar settings appear in personalized recommendation systems, auctions, online dating, political negotiations, finance, drug discovery, etc.
>
> ---
> > I also didn’t understand the importance of the “warm up” step on line 6 of the algorithm
>
> During the first $m-1$ episodes of each epoch, the adversary responds with possibly $m-1$ different strategies. From episode $m$ onward, the adversary responds with the same strategy, due to being memory-bounded by $m$. There’s nothing the learner can learn about the adversary during the first $m-1$ episodes but the $m^{th}$ episode onward. Thus, the data collected during the first $m-1$ episodes are not helpful for minimizing the policy regret, thus discarded.

---

### Decision · Program_Chairs · 2025-05-01

**Decision:**

Accept (poster)

**Comment:**

This paper studies the problem of policy regret minimization in Markov games that can handle function approximation (specifically, function classes with bounded Eluder dimension) and can deal with adaptive adversaries. Ultimately, all reviewers are positive about the paper's contributions, despite some limitations that were raised regarding the relevance on Eluder dimension and the lack of computational efficiency of the approach. Based on this feedback, I am recommending acceptance.